# Impact of vaccinations, boosters and lockdowns on COVID-19 waves in French Polynesia

Lloyd A. C. Chapman [1,2] ✉, Maite Aubry[3], Noémie Maset[4], Timothy W. Russell [1], Edward S. Knock [5], John A. Lees [5,6], Henri-Pierre Mallet[4], Van-Mai Cao-Lormeau [3] & Adam J. Kucharski[1,3]

Estimating the impact of vaccination and non-pharmaceutical interventions on COVID-19 incidence is complicated by several factors, including successive emergence of SARS-CoV-2 variants of concern and changing population immunity from vaccination and infection. We develop an age-structured multi-strain COVID-19 transmission model and inference framework to estimate vaccination and non-pharmaceutical intervention impact accounting for these factors. We apply this framework to COVID-19 waves in French Polynesia and estimate that the vaccination programme averted 34.8% (95% credible interval: 34.5–35.2%) of 223,000 symptomatic cases, 49.6% (48.7–50.5%) of 5830 hospitalisations and 64.2% (63.1–65.3%) of 1540 hospital deaths that would have occurred in a scenario without vaccination up to May 2022. We estimate the booster campaign contributed 4.5%, 1.9%, and 0.4% to overall reductions in cases, hospitalisations, and deaths. Our results suggest that removing lockdowns during the first two waves would have had non-linear effects on incidence by altering accumulation of population immunity. Our estimates of vaccination and booster impact differ from those for other countries due to differences in age structure, previous exposure levels and timing of variant introduction relative to vaccination, emphasising the importance of detailed analysis that accounts for these factors.

Since late 2020, multiple new severe acute respiratory coronavirus 2 (SARS-CoV-2) variants have emerged and spread globally, of which the major variant groups (Alpha, Beta, Gamma, Delta, and Omicron) have shown substantially different levels of transmissibility, severity and/or immune escape. At the same time, first- and second-dose vaccinations and booster doses against COVID-19 have been rolled out in many countries around the world, drastically changing population-level immunity and reducing incidence of severe COVID-19 outcomes[1–6].

Many countries have experienced multiple waves from the same or different variants[7–9]. In this context, estimating the impact of vaccination and non-pharmaceutical interventions (NPIs) on COVID-19 incidence is challenging, because it is necessary to account for: different variant properties, a complicated and ever-changing immune landscape from vaccination and previous infection, and the timing of variant emergence relative to vaccination roll-out and previous epidemic waves. Most existing modelling analyses of vaccination impact

[1]Centre for Mathematical Modelling of Infectious Diseases, London School of Hygiene and Tropical Medicine, London, UK. [2]Department of Mathematics and Statistics, Lancaster University, Lancaster, UK. [3]Laboratoire de recherche sur les infections virales émergentes, Institut Louis Malardé, Tahiti, French Polynesia. [4]Cellule Epi-surveillance Plateforme COVID-19, Tahiti, French Polynesia. [5]MRC Centre for Global Infectious Disease Analysis, School of Public Health, Imperial College London, London, UK. [6]European Molecular Biology Laboratory, European Bioinformatics Institute EMBL-EBI, Cambridgeshire, UK. ✉e-mail: l.chapman4@lancaster.ac.uk

have not explicitly accounted for different variant properties and the array of different levels and types of immunity that now exist[1–4,10], and thus may no longer offer the best available evidence. While frameworks for modelling multiple variants have been developed[11–15], most are country specific and not straightforwardly generalisable to other settings, or do not provide robust and flexible inference methodology for fitting to multiple data streams. Here we develop a framework that explicitly addresses these issues and apply it to COVID-19 epidemic waves in French Polynesia.

As of mid 2023, French Polynesia had experienced five waves of COVID-19 cases. The first, caused by the wild-type virus, started in August 2020 and peaked in early November 2020 (Fig. 1). Transmission then declined with the introduction of strict control measures, including a ban on gatherings in public places, mandatory mask wearing and a curfew, until cases reached very low levels again in February 2021. At this time a seroprevalence survey of 463 individuals on the main islands of Tahiti and Moorea was conducted to estimate the level of immunity in the population, and seroprevalence was estimated as 19.0% (95% confidence interval 15.5–22.9%). The low level of cases−driven by imports−was then maintained until mid-2021 when the rollout of the 1st and 2nd vaccine doses occurred. Following the introduction of the Delta variant in June 2021, the country experienced a second larger and sharper wave of cases, hospitalisations and deaths, with cases peaking in mid-August 2021. A lockdown was implemented with the establishment of a curfew and confinement at home on the main island groups (the Windward and Leeward Islands) in August 2021, and cases declined quickly back to low levels in November 2021. A second seroprevalence survey of 673 individuals on Tahiti was performed in November and December 2021, in which seroprevalence from natural infection was estimated as 57.7% (95% confidence interval 53.8−61.4%)[16]. The arrival of the Omicron BA.1/BA.2 variants in late December 2021 led to a relatively large third wave of cases, but fewer hospitalisations and deaths than in the previous waves, which coincided with the rollout of first booster doses. During the first trimester of 2022, incoming travellers were screened for infection at the border using PCR (polymerase chain reaction)/antigen tests. The third wave had largely subsided by April 2022. French Polynesia experienced a fourth wave of cases mainly caused by the Omicron BA.5 and BA.4 variants between June and September 2022[16] and a fifth wave mainly caused by the BQ.1.1 Omicron sub-variant in November and December 2022. During the third, fourth and fifth waves, no strong NPIs (curfews or case isolation) were implemented.

To understand how immunity and control measures shaped observed dynamics in French Polynesia, we fit an age-structured multistrain COVID-19 transmission model to reported case, hospitalisation and death data up to May 2022, as well as data from the two aforementioned seroprevalence surveys. We then use the fitted model to estimate the impact of NPIs and vaccination on numbers of COVID-19 cases, hospitalisations and deaths, and to estimate the immune status of the French Polynesian population.

## Results

### Model fit

The fit of the model to the overall numbers of confirmed cases, hospitalisations, and hospital deaths between July 2020 and May 2022 is shown in Fig. 2, and the fit of the model to the age-stratified numbers of cases, hospitalisations and deaths is shown in Figures S5 and S4, and to the age-stratified data from the seroprevalence surveys in Figure S6. The model reproduces the overall patterns in the data well, but underestimates hospital deaths among 60+ year-olds in the second wave. The estimated number of symptomatic cases over time corresponds closely to the numbers of confirmed cases during the three waves (Fig. 2), with an estimated reporting rate of 0.47 (95% credible interval (CI) 0.46–0.49), i.e. 47% of symptomatic cases having been reported.

### Impact of NPIs

We estimate the counterfactual impact that the lockdowns during the first two COVID-19 waves had on the numbers of symptomatic cases, hospitalisations and deaths in each wave and overall by simulating the model without the estimated reductions in the transmission rate corresponding to the lockdown periods in the first and second waves (Figure S10). We run 500 simulations with parameter values drawn from the posterior distribution of the parameters from the model fitting and compare the numbers of symptomatic cases, hospitalisations and hospital deaths to those in simulations with the estimated reductions in the transmission rate with lockdowns, to account for uncertainty in the estimated parameter values. This gives the results shown in Fig. 3 and Table S8.

The estimated overall numbers of symptomatic cases, hospitalisations and hospital deaths from fitting the model to the observed data

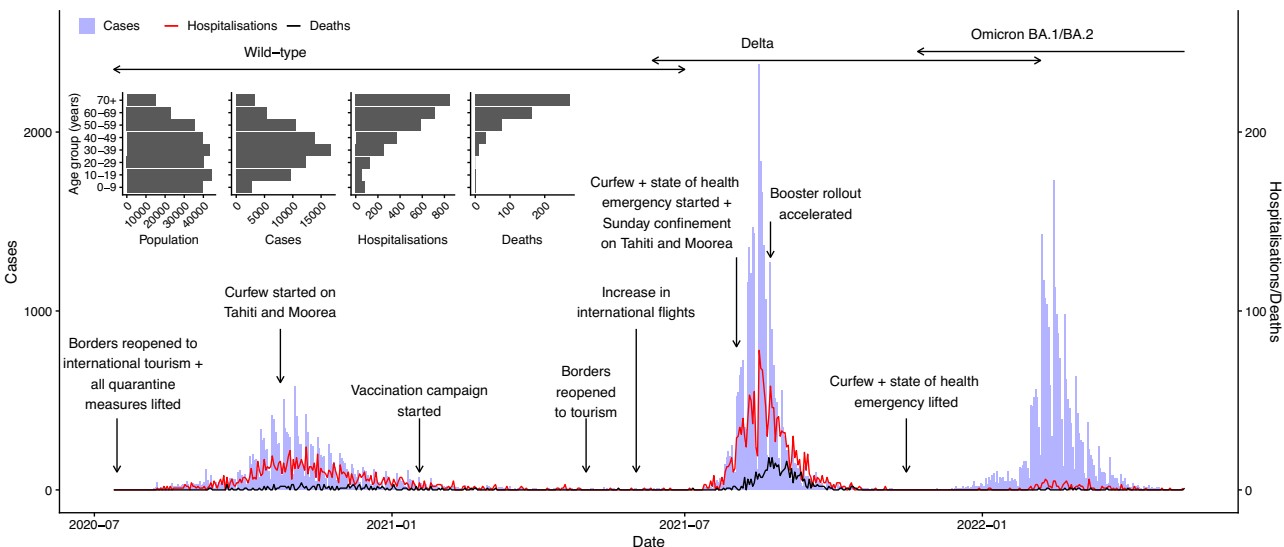

**Fig. 1 | COVID-19 epidemic in French Polynesia between August 2020 and May 2022.** Main panel: First three epidemic waves of COVID-19 confirmed cases, hospitalisations, and hospital deaths, with major changes in non-pharmaceutical interventions and vaccinations and boosters, and circulation periods of different variants. Inset: Age distributions of French Polynesian population, COVID-19 confirmed cases, hospitalisations and hospital deaths.

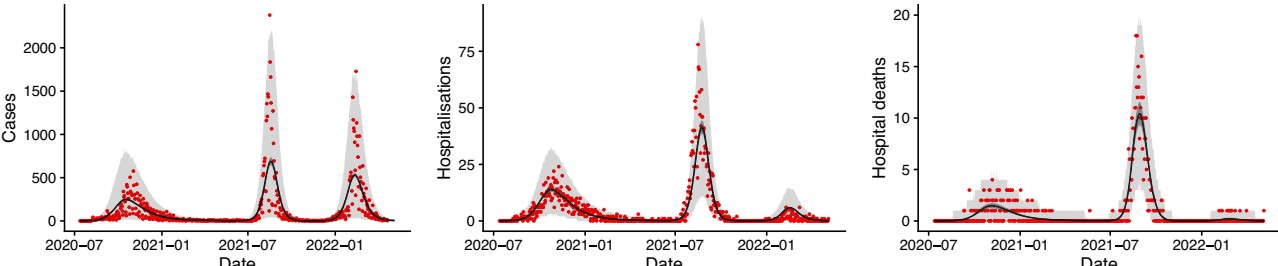

**Fig. 2 | Fit of the model to the observed total numbers of confirmed cases, hospitalisations and hospital deaths over time.** Red dots show observed counts, black line and dark grey shaded area show median and 95% CI of simulations of the fitted model, i.e. the uncertainty in the expected number of each outcome in the model. Light grey shaded area shows 95% posterior predictive interval of the model, i.e. the uncertainty in the values of each outcome from the model also accounting for uncertainty in the observation process. Note that there is a strong day-of-the-week effect in the reporting that accounts for the much of the dispersion in the data. Note different scales on vertical axes.

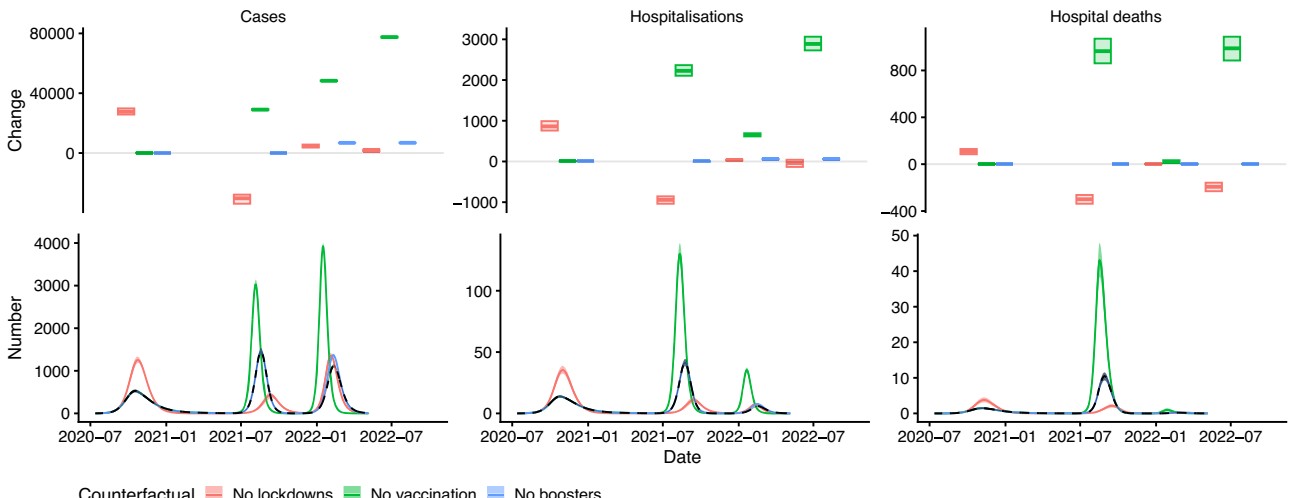

**Fig. 3 | Estimated impact of lockdowns, whole vaccination programme, and booster programme on numbers of COVID-19 symptomatic cases, hospitalisations, and hospital deaths.** Bottom panel: Numbers of cases, hospitalisations, and hospital deaths over time. Solid lines and shaded areas show medians and 95% CI of 500 simulations of the model for each counterfactual scenario, dashed black line and grey shaded area show median and 95% CI of simulations of the fitted model. Top panel: Change in cases, hospitalisations, and hospital deaths in each wave and overall (fourth set of boxes). Each box shows median (central line) and 95% CI (lower and upper lines) from 500 simulations of the model. See Figures S8 and S9 for results of sensitivity analysis.

up to May 2022 are 145,000 (95% CI 143,000–147,000), 2940 (95% CI 2810–3080) and 549 (95% CI 499–600) respectively. We estimated that removing the lockdowns in both the first and second waves would have had a non-linear effect on dynamics, and—assuming everything else had remained the same—would have led to fewer hospitalisations and hospital deaths over the study period from July 2020 to May 2022 (45 (95% CI -42–134) and 193 (95% CI 158–231) fewer, respectively) but a slightly higher number of symptomatic cases (1800 (95% CI 1300–2200) more). This scenario assumes that patients hospitalised during the first wave would have been managed the same (in terms of treatment and intensive care unit (ICU) admission) had the number of hospitalisations in the first wave been nearly 75% higher. The non-linear effect on overall incidence is due to the first wave of infections being much larger (with 27,600 (95% CI 25,700–29,900) more symptomatic cases, 860 (95% CI 760–990) more hospitalisations, and 105 (95% CI 85–129) more deaths), resulting in greater build up of immunity in the population prior to the introduction of the more severe Delta variant, and therefore a much smaller second wave of cases, hospitalisations and deaths (with 30,300 (95% CI 27,800–33,900) fewer cases, 940 (860–1040) fewer hospitalisations, and 299 (95% CI 263–338) fewer deaths). The impact on the third wave would have been relatively limited due to the effects on the first two waves approximately cancelling each other out in terms of cumulative infections, and the immune escape properties of the Omicron BA.1/ BA.2 variants reducing the influence of immunity from previous infection. Overall hospitalisations and deaths would have decreased, despite the increase in overall cases, as the reduction in cases in the second wave would have been slightly greater than the increase in cases in the first wave and there are more hospitalisations and deaths per case in the second wave than the first due to the greater severity of the Delta variant.

We also considered the counterfactual impact that changing the timings of the lockdowns during the first two waves would have had on incidence in each wave and overall (Supplementary Information §2.3). We estimated that starting the first and second lockdowns 2 weeks earlier or later would have had relatively little impact on overall numbers of cases, hospitalisations and deaths due to a similar non-linear cancellation effect between infections in the first and second waves as for removing the lockdowns (Figure S7).

## Impact of vaccination
The counterfactual impact of vaccination on numbers of hospitalisations and deaths during each wave and overall was estimated by simulating the fitted model without any vaccination, and comparing

the numbers of hospitalisations and deaths to those in simulations with the actual vaccination rollout (Fig. 3 and Table S8). The vaccination programme is estimated to have averted 77,500 (95% CI 77,200–77,800) symptomatic cases, 2890 (95% CI 2730–3070) hospitalisations and 989 (95% CI 885–1088) hospital deaths overall, with nearly all of these being averted during the second and third waves, since vaccination did not start until mid-January 2021 when the first wave had largely subsided. We also conducted a sensitivity analysis to determine the sensitivity of these estimates to uncertainty in the rates of waning of natural immunity and booster protection, cross-immunity to infection with Delta and Omicron from previous infection, and vaccine effectiveness (see Supplementary Information §§1.2 and 2.4 for details). The number of cases averted remains relatively constant across the range of parameter values considered, but the numbers of hospitalisations and deaths averted vary more significantly (from 2520 (95% CI 2380–2690) and 794 (95% CI 724–875) respectively under pessimistic assumptions about the parameters to 3630 (95% CI 3420–3860) and 1197 (95% CI 1080–1330) under optimistic assumptions).

Under our base case ('central') assumptions about the rate at which booster protection wanes, rate of waning of natural immunity, cross-immunity, and vaccine effectiveness, the booster campaign is estimated to have had a relatively small impact on the overall numbers of cases, hospitalisations and deaths, reducing them by 6800 (95% CI 6800–6900), 57 (95% CI 54–60) and 2 (95% CI 1–4) respectively (Fig. 3 and Table S8). However, these estimates are sensitive to uncertainty in these parameters. For pessimistic assumptions about the parameter values, the estimated reductions in cases, hospitalisations and deaths are 3300 (95% CI 3200–3300), 3 (95% CI 3–3), and 0 (95% CI 0–0) respectively, while for optimistic assumptions (including a less conservative assumption about the rate at which individuals lose all protection from boosters) they are 14,300 (95% CI 14,200–14,300), 163 (95% CI 153–173), and 9 (95% CI 6–15) respectively (see Supplementary Information §2.4 for further details).

**Immune status of the population**

The breakdown of the inferred immune status of the population over time and by age is shown in Fig. 4. The three waves of cases are visible where the proportion recovered from infection increases sharply in October 2020, August 2021, and February 2022. Based on the model, most infections in the Delta wave were among unvaccinated individuals without prior infection, while in the Omicron BA.1/BA.2 wave just under a half were among individuals with 2nd dose protection or waned 2nd dose protection, either with or without immunity from previous infection. The model also suggests that in May 2022 a very high proportion (78.9% (95% CI 78.5–79.5%)) of the population possessed either natural or hybrid (natural + vaccine-induced) immunity, and only 5.6% the population were fully susceptible. Table 1 shows the full breakdown of the estimated immune status of the population in May 2022. As expected, given prioritisation of older individuals in the vaccine and booster rollouts, the proportion of the population that had only natural immunity in May 2022 decreased with increasing age, from over 90% among 0-9-year-olds to approximately 16% among 70+ year-olds, while the proportion with hybrid immunity from infection and a booster dose increased with age from 0% among 0-9 year-olds to over 20% among 40+ year-olds.

## Discussion

Many existing modelling frameworks for estimating COVID-19 vaccination and NPI impact[1–4,10] do not explicitly account for the complicated COVID-19 immune landscape that now exists, with different levels of protection against different outcomes from different variants due to varying vaccination and infection histories and variant properties (transmissibility, immune escape, and severity). Those that do[11,13–15,17] tend to be country specific; reliant on detailed infection prevalence, hospitalisation and/or mobility data; and not readily transferable to estimate vaccination and NPI impact in settings without such data. We have developed an age-structured multi-strain SARS-CoV-2 transmission model that addresses this gap, and used it to estimate the impact of vaccination and non-pharmaceutical interventions on incidence of cases and severe outcomes in the first three waves of COVID-19 in French Polynesia. While our approach is similar to the 'stacked' SIR-type multistrain transmission model of different levels of immunity from vaccination and infection developed by LaJoie et al.[12], we also stratify by age and thus are able to model age-dependent mixing and infer variation in immunity over time by age group, and we use a more robust and flexible framework for performing parameter inference across multiple data sources rather than

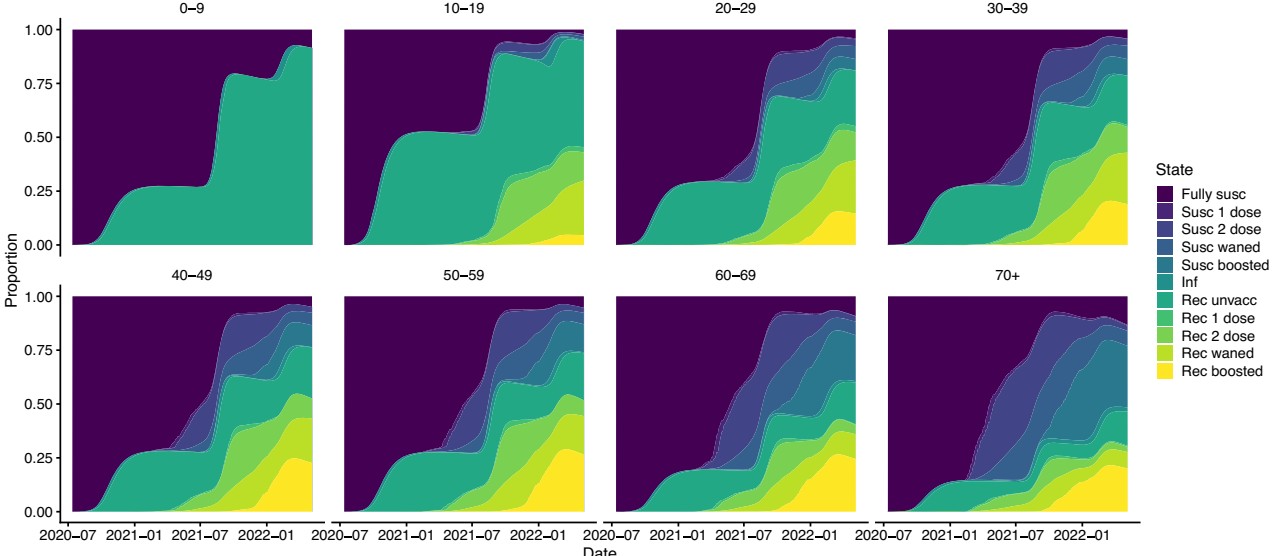

**Fig. 4 | Inferred immune status of the population over time by age group.** Coloured bands show median estimates from 1000 simulations of the fitted model. Note that the fully susceptible category includes individuals whose immunity from infection or vaccination has waned. Susc susceptible, Inf infected, Rec recovered.

**Table 1 | Estimated immune status of the overall population on 6th May 2022**

| State | Number, median (95% CI) | Percentage, median (95% CI) |
|---|---|---|
| Fully susceptible | 15800 (15400–16200) | 5.61 (5.47–5.76) |
| Susceptible 1 dose | 384 (374–395) | 0.137 (0.133–0.141) |
| Susceptible 2 dose | 5820 (5670–5980) | 2.07 (2.02–2.13) |
| Susceptible waned | 12700 (12400–13000) | 4.51 (4.4–4.63) |
| Susceptible boosted | 23500 (23100–24100) | 8.38 (8.22–8.57) |
| Infected | 841 (790–891) | 0.299 (0.281–0.317) |
| Recovered unvaccinated | 102000 (102000–103000) | 36.4 (36.3–36.6) |
| Recovered 1 dose | 2750 (2740–2760) | 0.979 (0.975–0.982) |
| Recovered 2 dose | 23800 (23600–23900) | 8.46 (8.4–8.51) |
| Recovered waned | 49800 (49500–50100) | 17.7 (17.6–17.8) |
| Recovered boosted | 43300 (42700–43700) | 15.4 (15.2–15.6) |

just reported cases. We have estimated impact through comparison with counterfactual scenarios, including no lockdowns, earlier/later introduction of lockdowns, no vaccination and no boosters. Our results suggest that the first two vaccine doses had a large impact on incidence in the Delta and Omicron BA.1/BA.2 waves, averting over 75,000 symptomatic cases, nearly 2900 hospitalisations and nearly 1000 deaths up to May 2022 compared to a counterfactual scenario of no vaccination.

Unlike many other Pacific Island Countries (PICs), which succeeded in preventing or delaying community transmission of SARS-CoV-2 till 2021 or till vaccine rollouts had started, French Polynesia experienced large-scale community transmission from relatively early in the pandemic in 2020[18]. Despite successful control of the wild-type virus through various NPIs, and the vaccine rollout occurring in mid-2021, the country suffered a very large wave of Delta infections and deaths in July–October 2021, with the highest COVID-19 death rate of any of the PICs. After the Delta wave, however, the level of immunity in the population had risen sufficiently to limit the burden of subsequent outbreaks in terms of hospitalisations and deaths, and French Polynesia experienced lower death rates than other PICs[19]. Nevertheless, the overall COVID-19 death rate in French Polynesia was still much higher than that of other PICs (231 deaths per 100,000 people vs the next highest of 110 for New Caledonia[20]). By providing a framework to quantify the impact of different interventions and evolution of population immunity in an island setting where local transmission becomes established, studies such as ours can offer valuable insight into intervention effectiveness and support planning of responses to future respiratory disease epidemics in such settings.

Although there have been global modelling analyses of vaccination impact at a country level[1,10], only one study has estimated vaccination impact in French Polynesia and our study is the first to estimate the impact in terms of different outcomes (cases, hospitalisations and deaths). A key difference between our study and these studies is that we fit our model to multiple direct data streams (hospital deaths, hospitalisations, reported cases and seroprevalence) rather than just estimates of excess deaths[21,22], which may make our estimates more robust. It does, however, lead to a much lower estimate of deaths averted through vaccination—Watson et al.[1] estimated that 2120 (95% CI 1740–2570) deaths had been averted up to 8th December 2021—since the number of hospital deaths with recorded date of death (552 between July 2020 and May 2022) is much lower than the estimated number of excess deaths (920 (95% CI 790–1200) between December 2020 and December 2021[22]). We chose to fit to hospital deaths rather than all deaths (hospital deaths + community deaths) since they are less sensitive to context bias as a data stream. We model variation in quality of patient care during the different waves by fitting a time-

dependent risk of death given hospitalisation. However, we do not account for changes in the risk of death in the community over the different waves, which may have been appreciable as there was considerable fear and distrust of hospitalisation during the Delta wave when hospitals reached capacity, and less distrust in healthcare in the first wave and less fear of severe outcomes during the Omicron BA.1/BA.2 wave. Given extensive follow-up of hospitalised cases it is likely that under-reporting of hospital deaths in French Polynesia was not as high as elsewhere[23].

We estimated that the booster campaign had less of an impact on the BA.1/BA.2 wave than the first two doses had on the Delta wave, in both absolute and proportional terms, despite similar numbers of infections in the BA.1/BA.2 wave as in the Delta wave. Although this may seem surprising, e.g. when compared with the estimated impact of the booster rollout in the UK[14], the estimated small effect size is influenced by a combination of factors. These include the already high level of natural/hybrid immunity in the population from previous infection and/or vaccination (Fig. 4), the relatively low booster coverage during the wave (< 35% of the overall population and only > 50% in individuals ≥50 years) (Fig. 5), and the lower severity of the BA.1/BA.2 variants.

Our results suggest that—all other things being equal—changing lockdown dates during the first two waves by two weeks would have had limited impact on overall numbers of cases, hospitalisations and deaths. This is because starting the first lockdown either earlier or later would have led to more infections prior to the second wave and thus been compensated for by a smaller second wave due to greater population immunity, and moving the second lockdown earlier or later would have had only a small impact on incidence due to its relatively limited estimated effect on the transmission rate (Figure S10). In addition, incidence in the Omicron BA.1/BA.2 wave would have been largely unaffected due to the limited net effect of changes in the lockdowns on incidence in the first two waves.

We also estimated the composition of immunity from infection and vaccination in the population in May 2022. We estimated that 94% of the population had some form of immunity, predominantly either natural or hybrid immunity (as opposed to only vaccine-induced immunity). From this we would expect that infection incidence (and therefore hospitalisation and death incidence) after May 2022 would have remained low without the advent of new variants with high levels of immune escape against Omicron BA.1/BA.2, which has been the case, with the Omicron BA.5/BA.4 and BQ.1.1 waves being relatively small in terms of detected cases, hospitalisations and deaths[24,25]. Given that the rollout of 2nd booster doses took place between April and August 2022 we would expect incidence of infections and severe outcomes to remain low for some time if no new immune escape variants are introduced.

There are some limitations to the analysis we have presented (see Supplementary Information for further discussion). Since there is no social contact data available for French Polynesia, we have to rely on contact data from the literature for the age-dependent contact rates used in the model and we choose to use data from France[26] (adjusted to account for French Polynesia's different population age structure). This is based on French Polynesia, as a French territory, having a societal structure more similar to that of France than most other countries and control measures implemented in French Polynesia during the pandemic being based on those in France.

We make the simplifying assumptions that mixing depends only on age and is otherwise homogeneous for the whole French Polynesian population, despite the fact that the population is spread over many islands in five archipelagos covering 2000km of ocean[16], and that the seroprevalence estimates from the main islands of Tahiti and Moorea are representative of seroprevalence on all the islands. However, the majority (~75%) of the population resides on Tahiti and Moorea (and ~69% on Tahiti), and most inhabited islands had frequent air

connections with Tahiti during the pandemic, except during lockdowns, so these assumptions are not unreasonable. The suspension of inter-island flights during lockdowns clearly would have had an impact on mixing of the population, but this is to some extent captured in the fitted values of the transmission rate parameters during the first and second epidemic waves. Ideally we would model transmission on the different archipelagos with a metapopulation model with the impact of lockdowns on inter-island movement informed by mobility data, or focus the analysis on the Windward Islands, which include Tahiti and Moorea, as these were most affected by the epidemic, but no mobility data is available for French Polynesia and we do not have sufficient geolocation detail for cases.

We assume that if lockdowns had been removed and hospitalisations had increased by nearly 75% during the first wave, hospital capacity would not have been exceeded and hospitalised patients would have received the same quality of care. Based on our estimates, the peak incidence of hospitalisations in the first wave would have been slightly lower than that that occurred during the second wave (Figure S9), when the number of general hospitalised and ICU COVID-19 cases at Centre Hospitalier de la Polynésie française (CHPF), the main hospital in French Polynesia where most COVID-19 patients were treated, peaked at 246 and 48 respectively. Since it was possible to make 248 general beds available for hospitalised cases at CHPF and ICU bed capacity there had already been upgraded to 36 in August 2020 (with the army placed on standby to set up 10 more ICU beds if required), it is therefore not unreasonable to assume hospitals would have remained within capacity. Nevertheless, the increased pressure on hospital resources might have led to lower quality of care for hospitalised patients and hence poorer outcomes. The estimated reduction in overall hospitalisations and hospital deaths from removing lockdowns should thus be interpreted with caution. Further, given the heterogeneity in type, compliance and duration of lockdowns between countries, and the relative uniqueness of French Polynesia in terms of remoteness and population size, this result is unlikely to be generalisable across countries.

We may underestimate the impact of the vaccination programme as we estimate cases, hospitalisations and deaths averted from reported hospital deaths, which are considerably lower than estimates of all-cause mortality and excess mortality[22,27], and do not account for increased death rates in the community when hospitals reached capacity during the Delta wave. The third wave was caused by a mixture of the Omicron BA.1 and BA.2 sublineages and there is evidence that the BA.2 variant is more transmissible than the BA.1 variant[28–31] and can reinfect individuals previously infected with the BA.1 variant[32], but we do not distinguish between these subvariants when modelling the third wave, which may lead to some underestimation of the impact of the booster programme.

We assume initial vaccine effectiveness and rates of waning of immunity are the same for all ages. However, there is evidence that vaccine effectiveness against symptomatic infection is lower and wanes more quickly in older age groups (≥65 years) than in younger

age groups, at least for the Delta variant[33], and of potential age differences in booster effectiveness against Omicron variants[34], which may introduce some bias into our estimates of vaccination and booster impact. We also assume waning rates are the same for different variants and infection outcomes, but data suggests waning is faster against the Omicron BA.1 variant than the Delta variant[35] and that protection against severe outcomes wanes more slowly than that against infection[33]. Further work is needed to determine the extent to which these differences affect vaccine impact estimates.

Nevertheless, the framework we have developed provides a means of estimating the impact of vaccination and NPIs on COVID-19 incidence while accounting for the complex immune landscape that has developed over the course of the pandemic from myriad different infection and vaccination histories at an individual level. In particular, several different data streams can be incorporated in the inference to provide more robust estimates of key unobserved processes. The framework is sufficiently flexible that it could be used to model COVID-19 dynamics and estimate vaccination and NPI impact for other countries, including those with less data available. As a minimum, COVID-19/excess death or hospitalisation data, case or seroprevalence data, vaccination and booster coverage data, dates of major changes in restrictions, and broad date ranges for the introduction of different variants would be required to fit the model, but in such a scenario all parameters except for the time-varying transmission rate, variant introduction dates, and symptomatic case reporting rate would need to be fixed to avoid parameter identifiability issues. In settings without seroprevalence data, case or regular testing data would be required to infer infection levels, or a fixed infection-hospitalisation/infection-fatality rate would have to be assumed. For countries with no variant sequencing data, date ranges for the introduction of different variants would have to be based on variant introduction dates for countries in that region or estimates of global emergence dates of new variants. Caution would be required to only apply the model to countries for which COVID-19 hospitalisation/death or excess death data was deemed to be reasonably complete to avoid biased estimates of impact. Nonetheless, the framework could still provide valuable insight in settings with different vaccine and booster availability and NPI levels.

## Methods

### Data

Multiple data streams are used in the fitting of the model. Anonymised line lists of confirmed cases and hospitalisations compiled by the Ministry of Health of French Polynesia with testing date and admission date, and date of death for those that died, and 10-year age group were aggregated into age-stratified time series of daily cases, hospitalisations and hospital deaths. Only 493 out of 74986 confirmed cases (0.66%) were missing their age group, so these cases were treated as unreported cases, since under-reporting of cases is accounted for in the model fitting (see *Confirmed cases*). As testing dates were missing for a large number of cases early in the first wave and cases were

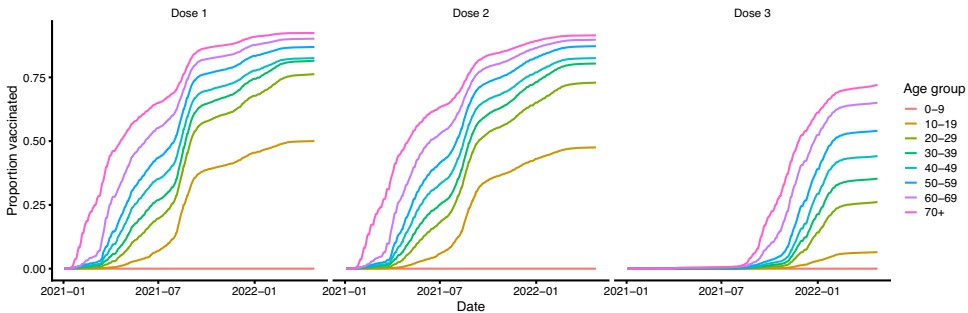

**Fig. 5 | Vaccination coverage by vaccine dose.** Dose 3 = booster dose.

**Table 2 | Seroprevalence in February 2021 and November-December 2021**

| Age group (years) | Feb 2021 survey | | | Nov-Dec 2021 survey | | |
|---|---|---|---|---|---|---|
| | Participants (n) | Seropositive (n) | Seroprevalence (%) (95% confidence interval*) | Participants (n) | Seropositive (n) | Seroprevalence (%) (95% confidence interval*) |
| 18–29 | 60 | 12 | 20 (10.8–32.3) | 169 | 103 | 60.9 (53.2–68.3) |
| 30–39 | 90 | 15 | 16.7 (9.6–26) | 163 | 105 | 64.4 (56.6–71.7) |
| 40–49 | 78 | 17 | 21.8 (13.2–32.6) | 92 | 54 | 58.7 (47.9–68.9) |
| 50–59 | 95 | 17 | 17.9 (10.8–27.1) | 79 | 48 | 60.8 (49.1–71.6) |
| 60–69 | 93 | 18 | 19.4 (11.9–28.9) | 113 | 56 | 49.6 (40–59.1) |
| 70+ | 47 | 9 | 19.1 (9.1–33.3) | 57 | 22 | 38.6 (26–52.4) |
| Total | 463 | 88 | 19.0 (15.5–22.9) | 673 | 388 | 57.7 (53.8–61.4) |

*95% confidence intervals are exact Clopper-Pearson binomial confidence intervals.

numbered approximately sequentially by testing date in the surveillance system, we imputed the missing dates as being between the testing dates of the nearest numbered cases with recorded testing dates. Data from two sero-surveys, the first conducted by Cellule Episurveillance COVID and the Health Department of French Polynesia in February 2021, the second by Institut Louis Malardé in November-December 2021, was also used. This data is described in detail in[16] and summarised in Table 2. Briefly, in February 2021, 463 unvaccinated adults aged 18–88 years on the islands of Tahiti and Moorea were randomly selected and tested for anti-SARS-CoV-2 immunoglobulin type G (IgG) antibodies with the Siemens SARS-CoV-2 IgG (sCOVG) test. Overall, 88 (19.0%, 95% confidence interval 15.5–22.9%) individuals had detectable IgG antibodies. In November-December 2021, 673 randomly selected individuals aged ≥18 years on Tahiti were tested for antibodies against the SARS-CoV-2 N antigen (i.e. for evidence of past infection) with the Roche Elecsys anti-SARS-CoV-2 assay, and 388 (57.7%, 95% confidence interval 53.8–61.4%) were positive. For the purposes of the modelling, we assume that the seroprevalence in the 20–29 years age group in the model is the same as that in the 18–29 years age group in the data. We use data on the population of French Polynesia by year of age in 2020 from the UN World Population Prospects[36] (for which the total population was estimated to be 280,904) aggregated into 10-year age groups for the age group populations in the model.

We use data on daily numbers of first, second and booster doses administered by age group (12–17, 18–29, 30–39, 40–49, 50–59, 60–69, 70+ years) collected by the Ministry of Health of French Polynesia to determine the numbers of individuals moving between the different vaccination strata in the model. Since the model is stratified into 10-year age groups, we split the doses in the 18-29 years age group in the data into the 10–19 and 20–29 age groups in the model according to population proportion (the proportions of 18–29 year-olds that are 18–19 and 20–29 years old). Upon division by the population in each age group, this gives the vaccination coverage by age and dose shown in Fig. 5.

**Model**

We developed a deterministic age-structured multi-strain SEIR-type model of COVID-19 transmission with stratification by vaccination status (Fig. 6). The model is stratified into 8 age groups (0–9, 10–19, 20–29, 30–39, 40–49, 50–59, 60–69, 70+ years), and by 5 vaccination levels representing no vaccination, protection from 1 dose, protection from 2 doses, waned protection from the 2nd dose and protection from a booster dose.

In the model, susceptible individuals (S) enter an exposed state (E) upon infection with a particular variant, from where an age-dependent proportion develop symptoms ($I_C$) after a presymptomatic infection period ($I_P$), while the rest progress to asymptomatic infection ($I_A$). Presymptomatic, symptomatic and asymptomatic individuals are all assumed to be infectious, though asymptomatic individuals less so. Most symptomatic individuals and all asymptomatic individuals recover naturally (R), but some symptomatic individuals develop severe disease (G or H) that can lead to hospitalisation. A proportion of these individuals die from the disease (D) while in hospital or at home, while the remainder recover following treatment. Infected individuals are assumed to cease being infectious upon recovery. Once recovered from infection individuals have immunity against reinfection with the same variant that wanes over time, but only partial immunity against infection with a different variant.

Individuals in the susceptible, exposed, presymptomatic, asymptomatic and recovered states can be vaccinated, providing them with increased levels of protection against infection, hospitalisation and death. The different vaccination strata and their associated levels of protection are shown in Tables 3 and S1.

The model is further stratified to account for different histories of infection with two different variants: the latest variant to have emerged and the previously dominant variant. Individuals can have been infected by only the previous variant, only the current variant, or the previous variant and the current variant (in either order), giving 4 possible infection histories. Once a new variant emerges the information stored in the strata for the two variants is combined into the stratum for the first variant, and the information for the new variant added to the second stratum. This simplification of the multistrain dynamics is to prevent the dimensionality of the model exploding as the number of variants and possible infection and vaccination histories increases, which would make the model prohibitively slow to fit.

Here we ignore transmission of the Alpha variant, as although Alpha was detected among travellers and a small number of local cases in early 2021 through variant screening (Table S5), transmission of Alpha remained localised and never became fully established. We therefore only explicitly model the introduction and spread of the Delta and Omicron variants. We also do not distinguish between the Omicron BA.1 and BA.2 sublineages, and model the introduction of Omicron and its sublineages as a single new variant.

Naturally-acquired immunity is assumed to wane slowly—individuals who have been infected are assumed to return to being susceptible to infection with the same variant after an exponentially distributed period with a mean of 6 years[14]. Immunity between SARS-CoV-2 variants is assumed to be asymmetric, with infection with later variants conferring stronger protection against infection with earlier variants than vice versa (see *Force of infection* and Table S2 for details). Changes in population-level serological status with seroconversion and seroreversion following infection are modelled with a 'parallel flow'.

Demographic processes such as birth, natural death and migration are ignored in the model (i.e. the population is assumed to remain constant in the absence of deaths from COVID-19). These processes occur at a much slower rate than transmission processes and are

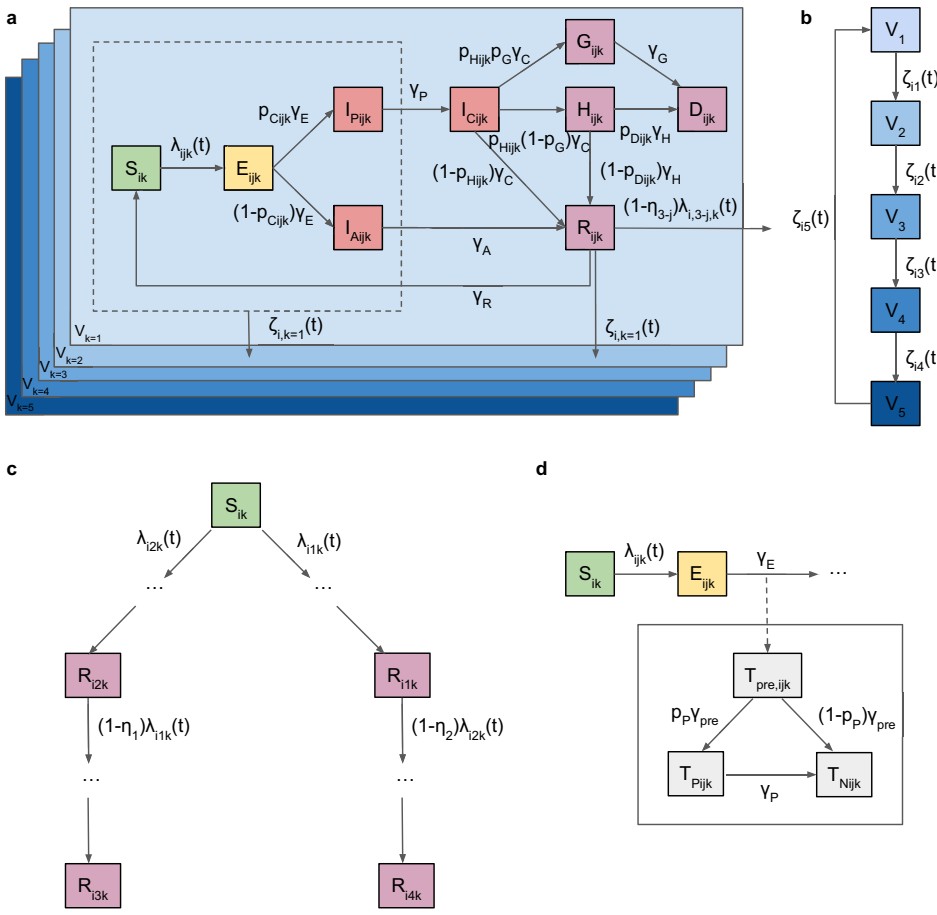

**Fig. 6 | Model flow diagram. a** SEIR-type transmission model structure with infectious states shown in red and different vaccination strata shown in blue. $S_{ik}$, $E_{ijk}$, $I_{Aijk}$, $I_{Pijk}$, $I_{Cijk}$, $H_{ijk}$, $G_{ijk}$, $D_{ijk}$, and $R_{ijk}$ denote the numbers of individuals who are susceptible, exposed (latently infected), asymptomatically infected, pre-symptomatically infected, symptomatically (clinically) infected, hospitalised, severely diseased who will die outside hospital, dead from COVID-19, and recovered from infection respectively. $T_{preijk}$, $T_{Pijk}$, and $T_{Nijk}$ denote the numbers of individuals pre-seropositive, seropositive, and seronegative against the SARS-CoV-2 N antigen. Subscripts denote the age group ($i \in$ {0–9, 10–19, 20–29, 30–39, 40–49, 50–59, 60–69, 70+} years), variant ($j \in$ {1, 2, 3, 4}, where $j = 3$ represents infection by variant 1 followed by infection by variant 2, and $j = 4$ vice versa), and vaccination stratum ($k \in$ {1, 2, 3, 4, 5}). Individuals in states inside dashed box and recovered from infection can move between vaccination strata upon vaccination. **b** Vaccination strata flow diagram (strata defined in Table 3). **c** Multi-strain model structure showing possible infection with first variant or second variant, or first then second, or second then first. **d** Seropositivity model structure with 'parallel flow' to transmission model flow. Transition rates between states are shown on arrows (see *Model equations* section and Table S2 for definitions). Further details of the model structure are provided in the *Methods* section.

**Table 3 | Vaccination strata in the model**

| Vaccination stratum | Dose number | Vaccine effectiveness | Mean duration | References |
|---|---|---|---|---|
| $V_1$ | 0 | None | Determined by vaccine roll-out | |
| $V_2$ | 1 | Full 1st dose effectiveness (28 days after 1st dose) | Determined by vaccine roll-out | |
| $V_3$ | 2 | Full 2nd dose effectiveness (14 days after 2nd dose) | 6 months | 37 |
| $V_4$ | 2 | Waned 2nd dose effectiveness | Determined by vaccine roll-out | |
| $V_5$ | 3 | Booster effectiveness | $-80/(365 \log(0.818)) = 1.1$ years | 35 |

therefore assumed to have a negligible impact on the transmission dynamics over the timescales modelled.

## Vaccination

Details of the five vaccination strata in the model are shown in Table 3. Unvaccinated individuals move out of the first vaccination stratum ($V_1$) into the first vaccinated stratum ($V_2$) at a rate determined by the roll-out of the 1st vaccine dose, with an assumed delay of 28 days for immunity from the 1st dose to develop. Likewise, movement into the 2nd dose vaccination stratum ($V_3$) is determined by the roll-out of the 2nd dose, with a delay of 14 days for the dose to take full effect. Only

non-symptomatic and non-hospitalised individuals, i.e. individuals in the $S$, $E$, $I_A$, $I_P$ and $R$ states in the model, can be vaccinated. Protection from the 2nd vaccine dose is assumed to wane over an exponentially distributed period, with a mean duration of 6 months. Individuals whose protection wanes pass into a 'waned' vaccine stratum ($V_4$), with lower levels of protection. They either remain in this stratum or receive a booster vaccination and move into a 'boosted' vaccination stratum ($V_5$), with higher levels of protection. Individuals can only move between consecutive vaccine strata except when they are in the 2nd dose stratum ($V_3$), where they can receive their booster dose and move to the boosted stratum ($V_5$) before their protection has waned,

skipping the waned 2nd dose stratum ($V_4$). Individuals in the 2nd dose and waned 2nd dose strata ($V_3$ and $V_4$) are taken to be equally likely to receive a booster dose. Protection from the booster dose is assumed to wane slowly such that individuals eventually return to being fully susceptible.

In common with other transmission modelling studies[14,37], we model vaccine protection against five different outcomes:

1. infection, with effectiveness $e^{inf}$
2. symptomatic infection given infection, $e^{sympt|inf}$
3. severe disease given symptomatic infection, $e^{SD|sympt}$
4. death given severe disease, $e^{death|SD}$
5. onward transmission if infected, $e^{ins}$

Vaccine effectiveness against symptomatic infection, severe disease and death are conditional on previous outcomes and depend on overall vaccine effectiveness against infection, symptomatic infection, severe disease and death ($e^{inf}$, $e^{sympt}$, $e^{SD}$ and $e^{death}$) as follows:

$$e^{sympt|inf} = \frac{e^{sympt} - e^{inf}}{1 - e^{inf}} \tag{1}$$

$$e^{SD|sympt} = \frac{e^{SD} - e^{sympt}}{(1 - e^{inf})(1 - e^{sympt|inf})} = \frac{e^{SD} - e^{sympt}}{1 - e^{sympt}} \tag{2}$$

$$e^{death|SD} = \frac{e^{death} - e^{SD}}{(1 - e^{inf})(1 - e^{sympt|inf})(1 - e^{SD|sympt})} = \frac{e^{death} - e^{SD}}{1 - e^{SD}} \tag{3}$$

Estimates for $e^{inf}$, $e^{sympt}$, $e^{SD}$ and $e^{death}$ for different vaccination strata and variants taken from[14] are provided in Table S1 (see[14] for information on sources of these estimates), but we also vary these parameters in the sensitivity analysis (Table S7). As over 90% of the doses given in French Polynesia were of the Pfizer-BioNTech vaccine, we use effectiveness values for that vaccine for all doses given. We also make the simplifying assumption that vaccine effectiveness is the same across all age groups.

### Waning immunity
The model accounts for waning of natural and vaccine-induced immunity as described in the previous sections. We assume that the waning rates of natural and vaccine-induced immunity are the same for all age groups and virus variants. When immunity from previous infection or booster vaccination wanes, individuals return to being fully susceptible, so immunity against different outcomes (infection, symptomatic infection, hospitalisation and death) is assumed to wane at the same rate. We note that this is a strong simplifying assumption as there is evidence to suggest that immunity against infection wanes more quickly than immunity against severe outcomes[33], and that immunity against infection and severe outcomes wanes faster for Omicron BA.1 than Delta[35]. As there is no data that provides a direct measure of the rate of loss of all protection from vaccination, we use the rate of waning of protection against hospitalisation as a proxy for the rate at which individuals return to being fully susceptible following booster vaccination. Whilst a reasonable assumption, this may still be overly conservative, so we also conduct a sensitivity analysis with different values of the waning rate from the literature. Although we do not model variant-specific vaccine waning rates, we use estimates of the change in protection against hospitalisation over time following booster administration for Omicron BA.1[35] for the booster waning rate in the analysis in the main text, since the booster campaign in French Polynesia coincided with the Omicron BA.1/BA.2 wave, and compare this to a lower

waning rate assumed by Barnard et al.[14] in their model with a similar structure (Table S6). See Supplementary Information §2.4 for results of the sensitivity analysis.

### Parallel flow for serological status
So that we can fit to the data from the sero-surveys we include a 'parallel flow' of compartments for serological status in addition to those for infection status and clinical progression (Fig. 6). We fit to the data on prevalence of seropositivity against the N antigen on the SARS-CoV-2 virus according to the Roche Elecsys anti-SARS-CoV-2 assay, as this tests only for positivity resulting from infection. After a pre-conversion period ($T_{pre}$), individuals either seroconvert ($T_P$) with probability $p_P$ or not ($T_N$). Those that do seroconvert eventually sero-revert (to $T_N$) after an exponentially distributed time with mean 6.6 years[38].

### Behaviour
The impact of lockdowns on transmission is described in the model through a time-varying transmission rate, with changepoints corresponding to major changes in restrictions in French Polynesia (Fig. 1). We make the simplifying assumption that adherence to these restrictions is the same across all age groups and vaccination strata, and regardless of infection history. Variation in care-seeking behaviour with age is modelled through an age-dependent probability of hospitalisation given symptomatic infection, where the relative risks of hospitalisation between age groups are based on data from France[39] and we estimate the maximum probability of hospitalisation across all age groups to account for differences in care-seeking and access to care between France and French Polynesia. The probability of hospitalisation varies across vaccination strata in the model due to the different levels of protection against severe disease with different levels of vaccination described above (see *Vaccination*), but we do not model any variation in care-seeking behaviour with vaccination status beyond this. Vaccine and booster uptake by age and vaccination status in the model are determined by the data on the numbers of each dose received by age over time (Fig. 5), assuming that all individuals within each age-and-vaccination stratum have an equal chance of being vaccinated (i.e. previous infection does not affect vaccine/booster-seeking) and can only receive successive vaccine doses (e.g. must have had the 2nd dose to receive a booster).

### Model equations
**Force of infection.** The relative susceptibility to infection with variant $j$ of a susceptible individual in age group $i$ in vaccination stratum $k$ is given by:

$$\chi_{ijk} = 1 - e_{ijk}^{inf}, \tag{4}$$

where $e_{ijk}^{inf}$ is the vaccine effectiveness against infection with variant $j$ in vaccination stratum $k \in \{1, 2, 3, 4, 5\}$ (see Table S1), and $\chi_{ij1} = 1$, $\forall\ i, j$ (i.e. there is no protection in unvaccinated individuals). The index $j$ denotes individuals' infection histories, covering primary infection with one variant ($j \in \{1, 2\}$) and superinfection (infection with one variant followed by infection with another) ($j \in \{3, 4\}$) as follows:

$$j = \begin{cases} 1 & \text{if individuals have only been infected by 1st variant,} \\ 2 & \text{if individuals have only been infected by 2nd variant,} \\ 3 & \text{if individuals infected by 1st variant followed by 2nd variant } (1 \rightarrow 2), \\ 4 & \text{if individuals infected by 2nd variant followed by 1st variant } (2 \rightarrow 1). \end{cases} \tag{5}$$

We describe two periods of the epidemic with this setup, the first running up to 21st November 2021 and encompassing the wild-type

and Delta waves, in which:

$$j = \begin{cases} 1 = Wildtype, \\ 2 = Delta, \\ 3 = Wildtype \rightarrow Delta, \\ 4 = Delta \rightarrow Wildtype. \end{cases} \quad (6)$$

and the second, starting on 21st November 2021 shortly before the emergence of Omicron BA.1 and ending on 6th May 2022 and covering the Omicron BA.1/BA.2 wave, in which:

$$j = \begin{cases} 1 = Delta, \\ 2 = Omicron, \\ 3 = Delta \rightarrow Omicron, \\ 4 = Omicron \rightarrow Delta. \end{cases} \quad (7)$$

The relative infectiousness of an individual in age group $i$ and vaccination stratum $k$ infected with variant $j$ compared with an unvaccinated individual infected with the wild-type virus is given by:

$$\xi_{ijk} = \sigma_j \left( 1 - e_{ijk}^{ins} \right) \quad (8)$$

where $\xi_{i,\text{Wildtype},1} = 1, \forall\, i$, and $\sigma_j$ is the relative transmissibility of variant $j$ compared to the wild-type variant (and we assume $\sigma_1 = \sigma_4$ and $\sigma_2 = \sigma_3$).

The infectiousness-weighted number of infectious individuals for variant $j$ in age group $i$ and vaccination stratum $k$ on day $t$ is given by

$$\Theta_{ijk}(t) = \xi_{ijk} \left( \theta_A I_{A,ijk} + I_{P,ijk} + I_{C,ijk} \right). \quad (9)$$

where $\theta_A$ is the relative infectiousness of an asymptomatic infected individual compared to a symptomatic individual in the same vaccination stratum infected with the same variant.

With these definitions, the force of infection on a susceptible individual in age group $i$ and vaccination stratum $k$ from variant $j$ on day $t$ is:

$$\lambda_{ijk}(t) = \begin{cases} \chi_{i1k} \sum_{i'} m_{ii'}(t) \sum_k (\Theta_{i',1,k}(t) + \Theta_{i',2\rightarrow1,k}(t)) & \text{if } j = 1, \\ \chi_{i2k} \sum_{i'} m_{ii'}(t) \sum_k (\Theta_{i',2,k}(t) + \Theta_{i',1\rightarrow2,k}(t)) & \text{if } j = 2. \end{cases} \quad (10)$$

where $m_{ii'}(t) = \beta(t) c_{ii'}$ is the time-varying person-to-person transmission rate from age group $i'$ to age group $i$, composed of the time-varying transmission rate $\beta(t)$ and the person-to-person contact matrix $c_{ii'}$ between age groups. The contact matrix $c_{ii'}$ was parameterised using estimates of contact rates $d_{ll'}^*$ between 5-year age groups for France from[26], where $d_{ll'}^*$ is the mean number of contacts in age group $l'$ an individual in age group $l$ makes per day. Following[26] and[40], these were corrected by the relative population densities of each age group of French Polynesia and France to account for differences in demography between the two countries:

$$d_{ll'} = d_{ll'}^* \frac{n_{l'}/n}{n_{l'}^*/n^*} \quad (11)$$

where $n_l^*$ and $n^*$ are the population of age group $l$ and the total population for France and $n_l$ and $n$ are those for French Polynesia. They were then averaged over 10-year age groups in the model and divided by the population in each age group to yield the person-to-person contact matrix $c_{ii'}$:

$$c_{ii'} = \frac{1}{n_{i'}} \frac{\sum_{l \in i} \sum_{l' \in i'} d_{ll'} n_l}{\sum_{l \in i} n_l}. \quad (12)$$

Social contact data for France was used due to the absence of estimates for French Polynesia and the fact that French Polynesia is a French territory.

The total force of infection on a susceptible individual in age group $i$ and vaccination stratum $k$ is then the sum of the variant-specific forces of infection:

$$\Lambda_{ik}(t) = \sum_{j=1}^{2} \lambda_{ijk}(t). \quad (13)$$

Cross-immunity between variants is modelled via partial immunity to infection with the other variant following infection with one variant, such that the force of infection on an individual recovered from infection with variant $j$ in age group $i$ and vaccination stratum $k$ from the other variant is:

$$\begin{cases} (1 - \eta_{3-j})\lambda_{i,3-j,k}(t) & \text{if } j \in \{1,2\}, \\ 0 & \text{if } j \in \{3,4\}, \end{cases} \quad (14)$$

where $\eta_j$ is the cross-immunity from infection with other variants against infection with variant $j$.

The time-varying transmission rate, $\beta(t)$, represents temporal changes in the overall contact rates in the population due to changes in restrictions and behaviour. We assume that $\beta(t)$ is piecewise linear with 5 changepoints corresponding to changes in alert levels and the imposition of island-wide restrictions such as curfews (Table 4 and Figure S10):

$$\beta(t) = \begin{cases} \beta_1 & \text{if } t \leq t_1 \\ \frac{t_i - t}{t_i - t_{i-1}} \beta_{i-1} + \frac{t - t_{i-1}}{t_i - t_{i-1}} \beta_i & \text{if } t_{i-1} < t \leq t_i, i \in \{2,\dots,5\} \\ \beta_5 & \text{if } t > t_5. \end{cases} \quad (15)$$

**Seeding of variants.** We seed each variant $j$ at a daily rate of $\omega_j$, over a period of $v_j$ days from time $t_j$. All seeding infections are from the S to E compartment in the 30-39-year-old age group and unvaccinated class.

For all variants, we seed at a rate of 10 infections per time step over one time step, i.e. $\omega_j = 40$ day$^{-1}$ and $v_j = 0.25$ days for $j \in \{Wildtype, Delta, Omicron\}$. We fit the seeding dates $t_0$ (which corresponds to the start date of the wild-type outbreak in 2020), $t_{Delta}$, and $t_{Omicron}$ (see Table 4).

The daily seeding rate of variant $j$ in age group $i$ in vaccine stratum $k$, $\delta_{ijk}(t)$, is therefore:

$$\delta_{ijk}(t) = \begin{cases} \omega_j & \text{if } i = [30,39), j \in \{Wildtype, Delta, Omicron\}, k = 0, t_j \leq t < t_j + v_j, \\ 0 & \text{otherwise}. \end{cases}$$

$$(16)$$

**Natural history parameters.** Movement between model compartments is determined by parameters $p_X$, defining the probability of progressing to compartment $X$, and rate parameters $\gamma_X$, defining the time individuals stay in compartment $X$, which can vary with age group ($i$), variant ($j$) and vaccination status ($k$). Values of these parameters are given in Tables S2 and S3 and information on how they are calculated is given below.

There is now strong evidence that successive SARS-CoV-2 variants have had progressively shorter serial intervals[41–44]. We therefore model this by reducing the mean durations of latent infection, asymptomatic infection, presymptomatic infection, and symptomatic infection ($E$, $I_P$, $I_C$, and $I_A$) of successive variants in line with percentage reductions in their serial intervals relative to the wild-type virus reported in the literature[42] (Table S4).

The probability of developing symptoms given infection is

$$p_{Cijk} = \left( 1 - e_{ijk}^{sympt|inf} \right) p_{Ci} \quad (17)$$

where $p_{Ci}$ is the age-dependent probability of developing symptoms given infection for unvaccinated individuals.

**Table 4 | Fitted model parameters prior and posterior distributions**

| Parameter | Description | Prior distribution | Posterior median (95% CI) |
|---|---|---|---|
| $\beta(t)$ | Transmission rate (per person) on day $t$ = YYYY-MM-DD | | |
| $\beta_1$ | 2020-08-27: Moved to level 3 (out of 4) alert, masking became obligatory | Gamma (4, 0.005) | 0.0207 (0.0204, 0.0211) |
| $\beta_2$ | 2020-10-24: Curfew established on the islands of Tahiti and Moorea | Gamma (4, 0.005) | 0.0157 (0.0154, 0.0159) |
| $\beta_3$ | 2021-06-01: Returned to level 1 alert, borders reopened, international flights increased | Gamma (4, 0.005) | 0.0204 (0.0197, 0.0210) |
| $\beta_4$ | 2021-08-02: Moved to stage 4 alert, before state of health emergency and curfew instigated | Gamma (4, 0.005) | 0.0181 (0.0175 0.0185) |
| $\beta_5$ | 2021-11-15: Returned to level 1 alert, curfew and state of health emergency lifted | Gamma (4, 0.005) | 0.0215 (0.0213, 0.0218) |
| $t_0$ | Start date of original outbreak | U[2020-07-01,2020-07-31] | 2020-07-03 (2020-07-01, 2020-07-06) |
| $t_{Delta}$ | Delta seeding date | U[2020-05-20,2020-06-29] | 2021-06-11 (2021-06-09, 2021-06-12) |
| $t_{Omicron}$ | Omicron seeding date | U[2021-11-21,2021-12-13] | 2021-11-22 (2021-11-21, 2021-11-23) |
| $p_{Hmax}$ | Maximum probability of severe disease requiring hospitalisation across all age groups | Beta (1, 1) | 0.233 (0.214, 0.255) |
| $p_{Dmax,1}$ | Maximum probability of death given hospitalisation across all age groups on (and before) 2021-06-11 | Beta (1, 1) | 0.224 (0.184, 0.268) |
| $p_{Dmax,2}$ | Maximum probability of death given hospitalisation across all age groups on 2021-08-15 | Beta (1, 1) | 0.748 (0.652, 0.846) |
| $p_{Dmax,3}$ | Maximum probability of death given hospitalisation across all age groups on (and after) 2021-11-01 | Beta(1, 1) | 0.0877 (0.0494, 0.142) |
| $\pi_{H\,Delta/Wildtype}$ | Relative risk of severe disease for Delta vs wild-type/Alpha | U (0,3) | 1.36 (1.21, 1.52) |
| $\phi_{cases}$ | Symptomatic case reporting rate | Beta (1, 1) | 0.474 (0.458, 0.492) |
| $\alpha_{cases}$ | Overdispersion parameter for negative binomial observation process for cases | Beta (1, 1) | 0.712 (0.672, 0.754) |
| $\alpha_{hosp}$ | Overdispersion parameter for negative binomial observation process for hospitalisations | Beta (1, 1) | 0.220 (0.168, 0.284) |
| $\alpha_{death}$ | Overdispersion parameter for negative binomial observation process for hospital deaths | Beta (1, 1) | 0.0475 (0.00211, 0.172) |

The probability that an individual develops severe disease requiring hospitalisation given that they are symptomatically infected is

$$p_{Hijk} = \left(1 - e_{ijk}^{SD|sympt}\right)\pi_{Hj}(1 - \eta_{Hj})p_{Hi} \qquad (18)$$

where $p_{Hi}$ is the age-dependent probability of developing severe disease given symptomatic infection for unvaccinated individuals, $\pi_{Hj}$ is the variant-dependent relative risk of severe disease, and $\eta_{Hj}$ is the cross-immunity from infection with other variants against hospitalisation with variant $j$. $p_{Hi}$ is defined as:

$$p_{Hi} = \psi_{Hi}p_{Hmax} \qquad (19)$$

where $p_{Hmax}$ is the maximum probability of hospitalisation across all age groups and $\psi_{Hi}$ is the age-dependent relative risk of severe disease, such that $\psi_{Hi} = 1$ for the group corresponding to the maximum. $\pi_{Hj}$ is parameterised as:

$$\pi_{Hj} = \begin{cases} \pi_{Delta/Wildtype} & \text{if } j = Delta, \\ \pi_{Delta/Wildtype}\pi_{Omicron/Delta} & \text{if } j = Omicron, \end{cases} \qquad (20)$$

where $\pi_{Delta/Wildtype}$ and $\pi_{Omicron/Delta}$ are the relative risks of severe disease given infection for Delta compared to wild-type and Omicron compared to Delta, and we fit $\pi_{Delta/Wildtype}$.

The probability that a hospitalised individual will die is

$$p_{Dijk}(t) = \left(1 - e_{ijk}^{death|SD}\right)\psi_{Di}(1 - \eta_{Dj})h(t) \qquad (21)$$

where $h(t)$ is the maximum probability of death given hospitalisation for unvaccinated individuals, $\psi_{Di}$ is the age-dependent relative risk of death for unvaccinated individuals (such that $\psi_{Di} = 1$ for the age group

for which the probability of death is $h(t)$), and $\eta_{Dj}$ is the cross-immunity from infection with other variants against death from variant $j$. To allow for variation in the risk of death with changing quality of care and demand for hospital beds, we fit a piecewise linear form for $h(t)$ with the following changepoints:

$$h(t) = \begin{cases} p_{Dmax,1} & \text{on (and before) 2021-06-11}, \\ p_{Dmax,2} & \text{on 2021-08-15}, \\ p_{Dmax,3} & \text{on (and after) 2021-11-01}, \end{cases} \qquad (22)$$

such that the probability of death given hospitalisation is constant during the first wave, changes with changing pressure on hospital beds in the Delta wave, and is constant after the Delta wave.

The probability that an individual dies in the community given that they have severe disease is

$$p_{Gijk} = \left(1 - e_{ijk}^{death|SD}\right)p_G \qquad (23)$$

where $p_G$ is the probability of death in the community given severe disease for unvaccinated individuals.

**Compartmental model equations.** The compartmental model is a deterministic approximation to a stochastic age-structured multi-strain SEIR-type transmission model in which draws from random variables are replaced by their expected values (using the deterministic mode of the `dust` R package). This may have lower accuracy than an ODE formulation and solver, but we expect that the error is minimal based on the model fits. The model compartments are defined in Fig. 6. For completeness we provide the equations for the stochastic model here, and note that the stochastic version of the model can be fitted and run by setting the option `deterministic <- F` in the code.

The compartments in the model are updated according to the following equations:

$$S_{ik}(t+dt) = S_{ik}(t) - \sum_{j=1}^{2} n_{SEijk} - \sum_{j=1}^{4} n_{seed,ijk} + n_{SVi,k-1} - n_{SVik} + \sum_{j=1}^{4} n_{RSijk} \quad (24)$$

$$E_{ijk}(t+dt) = E_{ijk}(t) + n_{SEijk} + \mathbb{1}_{j>2} n_{REi,j-2,k} - n_{EI_Aijk} - n_{EI_Pijk} + n_{EVij,k-1} - n_{EVijk} + n_{seed,ijk} \quad (25)$$

$$I_{Aijk}(t+dt) = I_{Aijk}(t) + n_{EI_Aijk} - n_{I_ARijk} + n_{I_AVij,k-1} - n_{I_AVijk} \quad (26)$$

$$I_{Pijk}(t+dt) = I_{Pijk}(t) + n_{EI_Pijk} - n_{I_PI_Cijk} + n_{I_PVij,k-1} - n_{I_PVijk} \quad (27)$$

$$I_{Cijk}(t+dt) = I_{Cijk}(t) + n_{I_PI_Cijk} - n_{I_CRijk} - n_{I_CHijk} - n_{I_CGijk} \quad (28)$$

$$H_{ijk}(t+dt) = H_{ijk}(t) + n_{I_CHijk} - n_{HRijk} - n_{HDijk} \quad (29)$$

$$G_{ijk}(t+dt) = G_{ijk}(t) + n_{I_CGijk} - n_{GDijk} \quad (30)$$

$$D_{ijk}(t+dt) = D_{ijk}(t) + n_{HDijk} + n_{GDijk} \quad (31)$$

$$R_{ijk}(t+dt) = R_{ijk}(t) + n_{I_ARijk} + n_{I_CRijk} + n_{HRijk} - n_{RSijk} - \mathbb{1}_{j\leq2} n_{REijk} + n_{RVij,k-1} - n_{RVijk} \quad (32)$$

$$T_{preijk}(t+dt) = T_{preijk}(t) + n_{EI_Aijk} + n_{EI_Pijk} - n_{T_{pre}T_Pijk} - n_{T_{pre}T_Nijk} \quad (33)$$

$$T_{Pijk}(t+dt) = T_{Pijk}(t) + n_{T_{pre}T_Pijk} - n_{T_PT_Nijk} \quad (34)$$

$$T_{Nijk}(t+dt) = T_{Nijk}(t) + n_{T_{pre}T_Nijk} + n_{T_PT_Nijk} \quad (35)$$

where $n_{XYijk}$ is the number of individuals in age group $i$ and vaccination stratum $k$ infected with variant $j$ (if they are in an infection state) moving from state $X$ to state $Y$ at time $t$ (and $n_{XYij0} = n_{XYij5}$, and we have dropped the dependence on $t$ from the notation for convenience); $dt$ is the model time step, chosen to be 0.25 days; and $\mathbb{1}_x$ is the indicator function for condition $x$.

The flows between states are determined as follows:

$$p_{SEijk} = \left(1 - e^{-\Lambda_{ik}(t)dt}\right)\frac{\lambda_{ijk}(t)}{\Lambda_{ik}(t)}, \quad j \in \{1,2\} \quad (36)$$

$$p_{SVik} = 1 - e^{-\zeta_{ik}(t)dt} \quad (37)$$

$$(n_{SEi1k}, n_{SEi2k}, n_{SSik}) \sim \text{Mult}\left(S_{ik}(t), p_{SEi1k}, p_{SEi2k}, 1 - \sum_{j=1}^{2} p_{SEijk}\right) \quad (38)$$

$$n_{seed,ijk} = \min\left(\text{Poiss}(\hat{\delta}_{ijk}(t)dt), S_{ik}(t) - \sum_{j=1}^{2} n_{SEijk}\right) \quad (39)$$

$$n_{SVik} = \text{Bin}\left(S_{ik}(t) - \sum_{j=1}^{2} n_{SEijk} - \sum_{j=1}^{4} n_{seed,ijk}, p_{SVik}\right) \quad (40)$$

$$p_{EI_Aijk} = (1 - p_{Cijk})\left(1 - e^{-\gamma_E dt}\right) \quad (41)$$

$$p_{EI_Pijk} = p_{Cijk}\left(1 - e^{-\gamma_E dt}\right) \quad (42)$$

$$p_{EVijk} = e^{-\gamma_E dt}\left(1 - e^{-\zeta_{ik}(t)dt}\right) \quad (43)$$

$$(n_{EI_Aijk}, n_{EI_Pijk}, n_{EVijk}, n_{EEijk}) \sim \text{Mult}\left(E_{ijk}(t), p_{EI_Aijk}, p_{EI_Pijk}, p_{EVijk}, 1 - \sum_{X\in\{I_A,I_P,V\}} p_{EXijk}\right) \quad (44)$$

$$(p_{I_ARijk}, p_{I_AVijk}) = \left(1 - e^{-\gamma_A dt}, e^{-\gamma_A dt}(1 - e^{-\zeta_{ik}(t)dt})\right) \quad (45)$$

$$(n_{I_ARijk}, n_{I_AVijk}, n_{I_AI_Aijk}) \sim \text{Mult}(I_{Aijk}(t), p_{I_ARijk}, p_{I_AVijk}, 1 - p_{I_ARijk} - p_{I_AVijk}) \quad (46)$$

$$(p_{I_PI_Cijk}, p_{I_PVijk}) \sim \left(1 - e^{-\gamma_P dt}, e^{-\gamma_P dt}(1 - e^{-\zeta_{ik}(t)dt})\right) \quad (47)$$

$$(n_{I_PI_Cijk}, n_{I_PVijk}, n_{I_PI_Pijk}) \sim \text{Mult}(I_{Pijk}(t), p_{I_PI_Cijk}, p_{I_PVijk}, 1 - p_{I_PI_Cijk} - p_{I_PVijk}) \quad (48)$$

$$p_{I_CHijk} = p_{Hijk}(1 - p_{Gijk})\left(1 - e^{-\gamma_H dt}\right) \quad (49)$$

$$p_{I_CGijk} = p_{Hijk}p_{Gijk}\left(1 - e^{-\gamma_H dt}\right) \quad (50)$$

$$p_{I_CRijk} = (1 - p_{Hijk})\left(1 - e^{-\gamma_H dt}\right) \quad (51)$$

$$(n_{I_CHijk}, n_{I_CGijk}, n_{I_CRijk}, n_{I_CI_Cijk}) \sim \text{Mult}\left(I_{Cijk}(t), p_{I_CHijk}, p_{I_CGijk}, p_{I_CRijk}, 1 - \sum_{X\in\{H,G,R\}} p_{I_CXijk}\right) \quad (52)$$

$$p_{HDijk} = p_{Dijk}\left(1 - e^{-\gamma_H dt}\right) \quad (53)$$

$$p_{HRijk} = (1 - p_{Dijk})\left(1 - e^{-\gamma_H dt}\right) \quad (54)$$

$$(n_{HDijk}, n_{HRijk}, n_{HHijk}) \sim \text{Mult}(H_{ijk}(t), p_{HDijk}, p_{HRijk}, 1 - p_{HDijk} - p_{HRijk}) \quad (55)$$

$$n_{GDijk} \sim \text{Bin}(G_{ijk}, 1 - e^{-\gamma_G dt}) \quad (56)$$

$$\gamma_{REijk} = \mathbb{1}_{j\leq2}(1 - \eta_{3-j})\lambda_{i,3-j,k} \quad (57)$$

$$p_{RSijk} = \left(1 - e^{-(\gamma_R + \gamma_{REijk})dt}\right)\frac{\gamma_R}{\gamma_R + \gamma_{REijk}} \quad (58)$$

$$p_{RE\,ijk} = \left(1 - e^{-(\gamma_R + \gamma_{RE\,ijk})dt}\right) \frac{\gamma_{RE\,ijk}}{\gamma_R + \gamma_{RE\,ijk}} \tag{59}$$

$$p_{RV\,ijk} = e^{-(\gamma_R + \gamma_{RE\,ijk})dt}\left(1 - e^{-\zeta_{ik}(t)dt}\right) \tag{60}$$

$$(n_{RS\,ijk}, n_{RE\,ijk}, n_{RV\,ijk}, n_{RR\,ijk}) = \text{Mult}\left(R_{ijk}(t), p_{RS\,ijk}, p_{RE\,ijk}, p_{RV\,ijk}, 1 - \sum_{X\in\{S,E,V\}} p_{RX\,ijk}\right) \tag{61}$$

$$p_{T_{pre}T_P\,ijk} = p_P\left(1 - e^{-\gamma_{pre}dt}\right) \tag{62}$$

$$p_{T_{pre}T_N\,ijk} = (1 - p_P)\left(1 - e^{-\gamma_{pre}dt}\right) \tag{63}$$

$$\begin{aligned}(n_{T_{pre}T_P\,ijk}, n_{T_{pre}T_N\,ijk}, n_{T_{pre}T_{pre}\,ijk}) \\ \sim \text{Mult}\left(T_{pre\,ijk}(t), p_{T_{pre}T_P\,ijk}, p_{T_{pre}T_N\,ijk}, 1 - p_{T_{pre}T_P\,ijk} - p_{T_{pre}T_N\,ijk}\right)\end{aligned} \tag{64}$$

$$n_{T_P T_N\,ijk} \sim \text{Bin}\left(T_{P\,ijk}(t), 1 - e^{-\gamma_P dt}\right) \tag{65}$$

where $n_{XX\,ijk}$ is the number of individuals in age group $i$ and vaccination stratum $k$ infected with variant $j$ (if they are in an infection state) who do not move from state $X$ at time $t$. The fitted seeding dates $t_0$, $t_{Delta}$, and $t_{Omicron}$ have continuous support, and the seeding process is handled within the discretisation to four update steps per day such that:

$$\hat{\delta}_{ijk}(t) = \begin{cases} \omega_j f_j(t) & \text{if } i = [30,39], j \in \{Delta, Omicron\}, k = 0, \\ 0 & \text{otherwise}. \end{cases} \tag{66}$$

where

$$f_j(t) = \begin{cases} \left\lceil \frac{t_j}{dt} \right\rceil - \frac{t_j}{dt} & \text{if } t = dt\left\lfloor \frac{t_j}{dt} \right\rfloor, \\ 1 & \text{if } dt\left\lfloor \frac{t_j}{dt} \right\rfloor < t < dt\left\lfloor \frac{t_j}{dt} \right\rfloor + \nu_j, \\ \frac{t_j}{dt} - \left\lfloor \frac{t_j}{dt} \right\rfloor & \text{if } t = dt\left\lfloor \frac{t_j}{dt} \right\rfloor + \nu_j, \\ 0 & \text{otherwise}. \end{cases} \tag{67}$$

## Model likelihood

The model likelihood is composed of the likelihoods for the different data streams that the model is fitted to, namely the age-stratified time series of hospitalisations, hospital deaths and confirmed cases, and the age-stratified seroprevalence data, as detailed below.

In the following, $Y \sim \text{Bin}(n,p)$ denotes that $Y$ follows a binomial distribution with $n$ trials and success probability $p$, such that

$$P(Y = y) = P_{\text{Bin}}(y|n,p) = \binom{n}{y} p^y (1-p)^{n-y}. \tag{68}$$

and the mean and variance of $Y$ are $np$ and $np(1-p)$ respectively. $Y \sim \text{NegBin}(m,\kappa)$ denotes that $Y$ follows a negative binomial distribution with mean $m$ and shape parameter $\kappa$, such that

$$P(Y = y) = P_{\text{NegBin}}(y|m,\kappa) = \frac{\Gamma(\kappa + y)}{y!\Gamma(\kappa)}\left(\frac{\kappa}{\kappa + m}\right)^\kappa \left(\frac{m}{\kappa + m}\right)^y \tag{69}$$

where $\Gamma(k)$ is the gamma function, and the variance of $Y$ is $m + m^2/\kappa$.

**Hospitalisations.** We assume that the observed number of hospitalisations in each age group $l$ at time $t$, $Y_{hosp,l}(t)$, is distributed according

to a negative binomial distribution

$$Y_{hosp,l}(t) \sim \text{NegBin}\left(X_{hosp,l}(t), \kappa_{hosp}\right) \tag{70}$$

with mean

$$X_{hosp,l}(t) = \sum_j \sum_k n_{I_C H\,ljk} \tag{71}$$

where the shape parameter $\kappa_{hosp}$ determines the overdispersion in the observation process and thus accounts for noise in the underlying data, and we aggregate the four youngest age groups together due to low numbers of hospitalisations in these age groups such that $l \in \{0{-}39, 40{-}49, 50{-}59, 60{-}69, 70+\}$ years. We fit the overdispersion parameter $\alpha_{hosp} = 1/\kappa_{hosp}$. The contribution of the age-stratified hospitalisation data to the likelihood is therefore:

$$L_{hosp} = \prod_t \prod_l P_{\text{NegBin}}(Y_{hosp,l}(t)|X_{hosp,l}(t), \kappa_{hosp}) \tag{72}$$

**Hospital deaths.** The observed number of hospital deaths in each age group $l \in \{0{-}39, 40{-}49, 50{-}59, 60{-}69, 70+\}$ years at time $t$ is assumed to be distributed according to a negative binomial distribution:

$$Y_{death,l}(t) \sim \text{NegBin}\left(X_{death,l}(t), \kappa_{death}\right) \tag{73}$$

with mean

$$X_{death,l}(t) = \sum_j \sum_k n_{HD\,ljk} \tag{74}$$

and shape parameter $\kappa_{death}$. We fit the overdispersion parameter $\alpha_{death} = 1/\kappa_{death}$. The contribution of the age-stratified hospital death data to the likelihood is thus:

$$L_{death} = \prod_t \prod_l P_{\text{NegBin}}(Y_{death,l}(t)|X_{death,l}(t), \kappa_{death}). \tag{75}$$

**Confirmed cases.** The daily number of confirmed cases in each age group $i \in \{0{-}9, 10{-}19, 20{-}29, 30{-}39, 40{-}49, 50{-}59, 60{-}69, 70+\}$ years is assumed to arise as the noisy under-reported observation of a hidden underlying Markov process

$$X_{cases,i}(t) = \sum_j \sum_k n_{EI_P\,ijk} \tag{76}$$

such that it follows a negative binomial distribution

$$Y_{cases,i}(t) \sim \text{NegBin}\left(\phi_{cases} X_{cases,i}(t), \kappa_{cases}\right) \tag{77}$$

with constant reporting factor $\phi_{cases}$ and shape parameter $\kappa_{cases} = 1/\alpha_{cases}$, where $\alpha_{cases}$ is an overdispersion parameter that we fit. The corresponding likelihood contribution is

$$L_{cases} = \prod_t \prod_i P(Y_{cases,i}(t)|X_{cases,i}(t), \kappa_{cases}, \phi_{cases}). \tag{78}$$

**Seroprevalence.** To fit the model to the age-stratified data from the two sero-surveys, we first calculate the number of seropositive and seronegative individuals in each age group over 20 years-of-age in the model (i.e assume the true serological status of all individuals is known):

$$X_{P_i}(t) = \sum_j \sum_k T_{P\,ijk}(t), \tag{79}$$

$$X_{N_i}(t) = N_i - \sum_j \sum_k T_{P\,ijk}(t), \quad i \in \{[20-29), \ldots, 70+\}. \tag{80}$$

We then compare the observed number of seropositive individuals in each age group in the sero-survey, $Y_{Pi}(t)$, with the number expected from the model based on the sample size $Y_{test,i}(t)$ and the sensitivity $p_{sens}$ and specificity $p_{spec}$ of the serological assay:

$$Y_{Pi}(t) \sim \text{Bin}(Y_{test,i}(t), \omega_P(t)) \tag{81}$$

where

$$\omega_{Pi}(t) = \frac{p_{sens}X_{Pi}(t) + (1 - p_{spec})X_{Ni}(t)}{X_{Pi}(t) + X_{Ni}(t)} \tag{82}$$

is the apparent prevalence. The likelihood contribution of the sero-survey data is:

$$L_{sero} = \prod_t \prod_i P_{\text{Bin}}(Y_{Pi}(t)|Y_{test,i}(t), \omega_{Pi}(t)) \tag{83}$$

**Full likelihood.** The full likelihood is the product of the likelihoods for the hospitalisation, death, case and sero-survey data:

$$L = L_{hosp}L_{death}L_{cases}L_{sero}. \tag{84}$$

## Prior distributions for fitted parameters

The prior distributions chosen for the fitted parameters are shown in Table 4. We use relatively informative gamma distributions for the transmission rate parameters $\beta_i \sim \text{Gamma}(k, \theta)$ ($i = 1, 2, 3, 4, 5$):

$$f(\beta_i) = \frac{1}{\Gamma(k)\theta^k} \beta_i^{k-1} e^{-\beta_i/\theta}, \quad x > 0, \tag{85}$$

where $\Gamma(\cdot)$ is the Gamma function, with shape parameter $k = 4$ and scale parameter $\theta = 0.005$ to ensure that the basic reproduction number for the wild-type variant is in a sensible range. Targeted sequencing of samples from local cases and travellers to screen for new variants was performed from late December 2020 in French Polynesia (Table S5). Whilst this data is biased and so cannot be used to fit the variant proportions in the model, it can be used to constrain the introduction dates of the different variants. We use continuous uniform prior distributions for the introduction dates of the different variants, with the upper bounds of the distributions for Delta and Omicron BA.1 chosen to match the earliest date each variant was detected amongst local cases (since the variant cannot have been introduced into local circulation later than it was first detected), and the lower bounds chosen as 40 days and 12 days earlier respectively based on the earliest date each variant was detected amongst travellers and the much higher growth rate of the Omicron BA.1 variant (Tables 4 and S5). For the wild-type variant, we assume a lower bound of 39 days prior to the first reported hospitalisation and an upper bound of 9 days prior. We treat the introduction dates as continuous variables, and distribute the initial number of infections of that variant in proportion to how far between time steps the introduction date is. This helps to avoid mixing issues in the MCMC caused by treating the introduction date as a discrete variable. For the maximum probability of severe disease across all age groups and the symptomatic case reporting rate, we use completely uninformative priors, $p_{Hmax}, \phi_{cases} \sim \text{Beta}(1,1)$, where the density for $X \sim \text{Beta}(a, b)$ is:

$$f(x) = \frac{\Gamma(a + b)}{\Gamma(a)\Gamma(b)} x^{a-1}(1 - x)^{b-1}, \quad x \in (0.1). \tag{86}$$

## MCMC algorithm

We use the accelerated shaping and scaling adaptive Markov Chain Monte Carlo (MCMC) algorithm of Spencer[45] to infer the values of the fitted parameters $\theta = (\beta, t_0, t_{Delta}, t_{Omicron}, p_{Hmax}, p_{Dmax,1}, p_{Dmax,2},$

$p_{Dmax,3}, \pi_{HDelta/Wildtype}, \phi_{cases}, \alpha_{cases}, \alpha_{hosp}, \alpha_{death})$, where $\beta = (\beta_1, \beta_2, \beta_3, \beta_4, \beta_5)$. The algorithm adaptively shapes and scales the proposal matrix to achieve more efficient mixing. We refer the reader to[45] for full details. The algorithm proceeds by repeating the following steps:

1. At the $i$th iteration, draw new values of the fitted parameters from a multivariate normal proposal distribution

$$\theta_i \sim N(\theta_{i-1}, 2.38^2 c_{i-1}^2 \Sigma_{i-1}/n_\theta) \tag{87}$$

where $\Sigma_{i-1}$ is the running estimate of the covariance matrix of the posterior distribution, $n_\theta$ is the dimension of the posterior density, and $c_{i-1}$ is a scaling parameter that is tuned to achieve a desired acceptance rate (see Step 4).

2. Accept $\theta_i$ with probability:

$$\alpha(\theta_i, \theta_{i-1}) = \min\left(1, \frac{L(\theta_i)P(\theta_i)}{L(\theta_{i-1})P(\theta_{i-1})}\right) \tag{88}$$

where $P(\theta)$ is the prior density of $\theta$.

3. Calculate the running mean and covariance as: if $i = 1$:

$$\bar{\theta}_1 = \frac{1}{2} \sum_{j=0}^1 \theta_j \tag{89}$$

$$\Sigma_1 = \frac{1}{i_0 + n_\theta + 3} \left((i_0 + n_\theta + 1)\Sigma_0 + \sum_{j=0}^1 \theta_j\theta_j^T - 2\bar{\theta}_1\bar{\theta}_1^T\right) \tag{90}$$

if $f(i) = f(i-1) + 1$, where $f(i) = \lfloor \frac{i}{2} \rfloor$:

$$\bar{\theta}_i = \bar{\theta}_{i-1} + \frac{1}{i - f(i) + 1}(\theta_i - \theta_{f(i)-1}) \tag{91}$$

$$\Sigma_i = \Sigma_{i-1} + \frac{1}{i - f(i) + i_0 + n_\theta + 2}\left(\theta_i\theta_i^T - \theta_{f(i)-1}\theta_{f(i)-1}^T\right.$$
$$\left. - (i - f(i) + 1)(\bar{\theta}_{i-1}\bar{\theta}_{i-1}^T - \bar{\theta}_i\bar{\theta}_i^T)\right) \tag{92}$$

such that the new observation replaces the oldest, and if $f(i) = f(i-1)$:

$$\bar{\theta}_i = \frac{1}{i - f(i) + 1}((i - f(i))\bar{\theta}_{i-1} + \theta_i) \tag{93}$$

$$\Sigma_i = \frac{1}{i - f(i) + i_0 + n_\theta + 2}\left((i - f(i) + i_0 + n_\theta + 1)\Sigma_{i-1} + \theta_i\theta_i^T\right.$$
$$\left. - (i - f(i))\bar{\theta}_{i-1}\bar{\theta}_{i-1}^T - (i - f(i) + 1)\bar{\theta}_i\bar{\theta}_i^T\right) \tag{94}$$

such that a new observation is included, where $i_0$ is a constant that determines the rate at which the influence of $\Sigma_0$ on $\Sigma_i$ decreases.

4. Update the covariance scaling parameter $c_i$:

$$c_i = \max\left(c_{min}, c_{i-1}\exp\left(\frac{\delta}{i_{start} + i}(\alpha(\theta_i, \theta_{i-1}) - a)\right)\right) \tag{95}$$

where

$$\delta = \left(1 - \frac{1}{n_\theta}\right)\frac{\sqrt{2\pi}\exp(A^2/2)}{2A} + \frac{1}{n_\theta a(1 - a)} \tag{96}$$

$$A = -\Phi^{-1}(a/2) \tag{97}$$

$$i_{start} = \frac{5}{a(1 - a)} \tag{98}$$

with $\Phi(\cdot)$ the cumulative distribution function of the standard normal distribution, $c_{min}$ is a minimum value for the scaling parameter (to prevent the proposal matrix being shrunk too much, which can lead to very slow mixing), and $a$ is the target acceptance rate.

5. If $|\log(c_i) - \log(c_{start})| > \log(3)$, restart the tuning of the scaling parameter from its current value:

$$c_{start} \mapsto c_i \qquad (99)$$

$$i_{start} \mapsto \frac{5}{a(1-a)} - i. \qquad (100)$$

We run 4 chains of the above algorithm from different initial parameter values with $i_0 = 100$, $c_0 = c_{start} = 1$, $c_{min} = 1$, and a target acceptance rate of $a = 0.234$ for 50,000 iterations. We thin the chains by a factor of 10, then discard the first 4000 iterations of each thinned chain as burn-in, and combine the remaining iterations to form a sample of size 4000. We assess convergence of the MCMC chains by visual assessment of the trace plots, and calculating the maximum Gelman-Rubin statistic and minimum effective sample size across all the parameters for the thinned combined sample (4000 iterations).

### Reporting summary
Further information on research design is available in the Nature Portfolio Reporting Summary linked to this article.

## Data availability
All data used in the analysis are available online on Zenodo at https://doi.org/10.5281/zenodo.8320333 and on GitHub at https://github.com/LloydChapman/covid_multi_strain.

## Code availability
The code used in this analysis was developed in R version 4.1.0 and uses the `odin`, `odin.dust`, `dust` and `mcstate` R packages for simulating discrete-time stochastic processes[46–49]. The model structure is similar to that of the COVID-19 transmission model in the `sircovid` R package[50], and some of the code from this package is reused. All code used in the analysis is available online at https://github.com/LloydChapman/covid_multi_strain.

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

## Acknowledgements

We would like to thank Richard FitzJohn of the MRC Centre for Global Infectious Disease Analaysis at Imperial College for help with the R package mcstate and Rosanna Barnard of the London School of Hygiene and Tropical Medicine for helpful discussions. LACC, ESK and AJK were supported by funding from the National Institute for Health and Care Research (NIHR) Health Protection Research Unit in Modelling and Health Economics, a partnership between the UK Health Security Agency (UKHSA), Imperial College London, and the London School of Hygiene & Tropical Medicine (NIHR200908). LACC and AJK were also supported by funding from the Wellcome Trust (206250/Z/17/Z). The views expressed are those of the authors and not necessarily those of the UK Department of Health and Social Care, NIHR, UKHSA, or Wellcome Trust.

## Author contributions

L.A.C.C., V.M.C.L., and A.J.K. conceived the study. L.A.C.C. wrote the computer code, conducted the analysis and wrote the first draft of the manuscript. M.A., N.M., and H.P.M. processed the data. T.W.R., E.S.K., J.A.L. provided support with the development of the code. L.A.C.C., M.A., H.P.M., V.M.C.L., A.J.K. interpreted the results and critically revised the manuscript. All authors read and approved the final version of the manuscript.

## Competing interests

The authors declare no competing interests.

## Ethics approval

Secondary data analysis of routinely collected COVID-19 data from French Polynesia was approved by the London School of Hygiene and Tropical Medicine Observational Research Ethics Committee (ref 28129).

## Additional information

**Peer review information** *Nature Communications* thanks Jamie Cald-well, Laura Skrip and the other, anonymous, reviewer(s) for their con-tribution to the peer review of this work. A peer review file is available.

