## [Peer Review File · Nature Communications]

Impact of vaccinations, boosters and lockdowns on COVID-19 waves in French PolynesiaREVIEWER COMMENTS

Reviewer #1 (Remarks to the Author):

Due to the small scale of the study and the topic, which has been explored at length for numerous countries and contexts (being of great relevance in 2021 and 2022 for COVID control), the results are not in themselves noteworthy in novelty or applicable in terms of policymaking. Global studies which have explored the questions posed by this paper have already been conducted (e.g. Watson et al 2022; [https://www.thelancet.com/journals/laninf/article/PIIS1473-3099\(22\)00320-6/fulltext](https://www.thelancet.com/journals/laninf/article/PIIS1473-3099(22)00320-6/fulltext), Moore et al 2021; [https://www.thelancet.com/journals/laninf/article/PIIS1473-3099\(21\)00143-2/fulltext](https://www.thelancet.com/journals/laninf/article/PIIS1473-3099(21)00143-2/fulltext))

Although the methodology is overall sound as separate components (except for potential notation issues identified below), the use of the SEIR (or similar frameworks) modelling to address this questions has been done repeatedly and is not novel, nor does it contribute significantly to our understanding of COVID-19 control or vaccination at this time. Examples include Aruffo et al. 2022; <https://bmcpublichealth.biomedcentral.com/articles/10.1186/s12889-022-13597-9> and Li et al. 2021; <https://www.frontiersin.org/articles/10.3389/frai.2021.648579/full>). Therefore I paid more attention to other aspects of the paper including the parallel flow, but found it challenging to accept the results due to the high complexity added for some components, and gross simplification of others.

It should also be noted that many countries experienced multiple waves of the same or differing variants (e.g. Screen et al 2022; https://wwwnc.cdc.gov/eid/article/28/13/22-0228_article) and have created models to explore this (e.g. LaJoie et al 2022; <https://www.nature.com/articles/s41598-022-24967-z>)

The complexity of the model also makes interpretation and disentangling the impact of each component challenging, also unfortunately leaving many uncertainties due to the numerous assumptions and simplifications built one on top of the other.

For example, the assumption that Omicron and all sublineages are a single variant and that susceptibility to the same variant infection has a mean of 6 years when data shows the relationship is incredibly complex across and within the same variants with estimates of much smaller time intervals of reinfection existing (COVID-19 Forecasting Team 2023; [https://www.thelancet.com/journals/lancet/article/PIIS0140-6736\(22\)02465-5/fulltext](https://www.thelancet.com/journals/lancet/article/PIIS0140-6736(22)02465-5/fulltext)). Thus I would expect this to be incorporated in terms of the uncertainty as a different distribution could cause the results to be very different in understanding the underlying immunity of the population.

The authors also state they used social contact data from France for the island states of French Polynesia, when the social dynamics are very different. The use of movement data or other social contact estimates from literature would be more appropriate here. The dynamics of spread would also be expected to differ substantially. They do state the limitation in the fact that these are islands - I think this cannot be ignored as movement restrictions with natural barriers in place would reduce spread drastically under lockdown conditions. This aspect of the study actually merits investigation and presents a more unique angle in comparison to the numerous homogeneous landscape studies which are available.

The impact of interventions also has received much attention as aforementioned. Disentangling the impact of lockdown in itself is a highly complex modelling exercise with considerations on compliance by location, heterogeneity in impact across age, sources of infection still occurring within the community/home/workplaces etc. The addition of vaccination via a parallel flow with different vaccination pathways (i.e. variant 1 to 2, variant 2 to 1) ignores the complexity of interventions, and instead adds substantial detail in infection histories which is difficult to separate from intervention effectiveness with their model framework. An agent based model may be more appropriate here if such detail is to be modelled, which allows full exploration of all parameter space, acknowledgment of the complexity of human behaviour through simulation - this would be reflected in the wider confidence intervals in the results which should exist considering both the complexity and the amount of parameter/distribution uncertainty which should exist in many of the parameters.

Whilst there is sufficient detail to repeat the exercise, the numerous assumptions would make the results difficult to stand by, especially for other contexts with larger populations and complex mixing, with longer periods of overlapping variants outbreaks or changing compliances to NPIs.

Minor issues:

A few citations are missing e.g. page 7, page 12

Might be wrong in page 14, Last two formulas on page 14, is it $(1 - e^{-\dots})$... rather than $e^{-\dots}$?

For the population $J = 3, 4$, the author assumed them to be waning in the immunity and removed from R population but did not add them back in the S population as the formula accepts only $J = 1, 2$ from R population - is there a notation issue?

Confusion on reasoning for combining the variant 1 and 2 into one stratum (henceforth variant 1) when variant 3 is introduced (which is henceforth denoted as variant 2)?

Reviewer #2 (Remarks to the Author):

Chapman et al. use an age-structured multi-strain COVID-19 transmission modeling framework to account for the impact of changes in underlying immunity on effectiveness of interventions, including vaccination and social distancing via lockdown. This is an important contribution to mathematical modeling of COVID-19 and the application to French Polynesia offers insights into the role vaccine-induced and exposure-based immunity have played. The manuscript is well written with detailed description of the model and data used to inform it. Suggestions below reflect model assumptions and findings that could warrant additional justification/comment.

Introduction:

- It would be helpful to have the seroprevalence study results (briefly) mentioned in the second paragraph to reflect seroprevalence within the timeline of the variants/waves/interventions.

Results:

- Paragraph 2: Removal of lockdowns in the first and second waves was associated with overall reduction in hospitalizations and hospital deaths. It is understood that this means across the entire period under study (768 and 107). However, the authors then suggest that the same scenario led to reductions in wave 2 that were greater than the reductions overall (1430 vs 768 and 230 vs 107). Please revisit the sentences to reword if needed.

Methods:

- Thank you for the clear variant-specific vaccination data in Table 5. It appears that vaccination parameters on effectiveness and waning were assumed to not be age-specific. Is this decision based on literature or a simplifying assumption? Could the authors please support the decision with references if the former or note the decision in the limitations if the latter. (Note evidence on potential age differences in booster effectiveness - <https://www.ecdc.europa.eu/en/publications-data/interim-analysis-covid-19-vaccine-effectiveness-against-severe-acute-respiratory>)

- Assumptions around contact patterns, care-seeking behavior, and 'effectiveness' of hospital care (probability of death among hospitalized cases) are that French Polynesia is similar to France. It is understood that French Polynesia is a French territory. However, could data from other settings (perhaps island settings) be more representative of the situation in French Polynesia? Or are there any thoughts on how sensitive the model is to these assumptions that France is representative of French Polynesia? Please consider including a discussion of this.

- It is noted that the authors fit a time-varying transmission parameter to account for impact of lockdown and use probability of hospitalization for care-seeking (based on data from France). However, given the focus of NPIs and vaccination in the paper, please consider explicitly adding a section of text to the Methods to account for how behavior parameters (care-seeking, vaccine/booster-seeking, and lockdown adherence) are addressed in the model, particularly across age and vaccination strata, and natural immunity compartments.

Discussion:

- The authors' work suggests that lockdowns rendered the situation worse for waves 2 and 3. This is understood in the context of immunity. However, could they better contextualize this in terms of

country-level resources for response? More specifically the authors assumed sufficient and uniform capacity for care despite increased case counts. Please consider commenting on the number of available hospital beds and whether the increase in case counts during the first wave would have exceeded this capacity. I see a note in the Supplement, but it seems like an important point (often contentious, multifaceted considerations that go into lockdown decisions) that may warrant attention in the Main Text, particularly as decision-makers may look to this evidence for response decisions in future/non-COVID outbreaks.

Supplement:

- Thank you for the extended discussion of Limitations. Could the authors also comment on how flexible the model is to address dynamics in other settings? Particularly, in settings where no seroprevalence data are available and/or less information is available on distributions of variants in the population. The model could provide valuable explanatory information for settings where lockdowns were less possible/enforced and vaccines (particularly boosters) have been less available, but these settings also have less data available. What minimum country-specific information is needed to generalize the model to other settings? Please consider discussing whether there should be any caution in using the model in a different context.
- In the Main Text, the authors note that "The model reproduces the overall patterns in the data, although it does not fully capture the flatness of the first wave of hospitalisations and hospital deaths, underestimates deaths among 60+ year-olds in the second wave, and overestimates hospitalisations in the third wave." Do any of the limitations presented here (or in the Discussion) address these observed variances? If so, please comment in the text.

Figures:

Figure 3 on relative immune statuses over time is very helpful to see. Well done.

Figure 5 legend – There seem to be five vaccination strata: $k=1$ through $k=5$. Please check the subscript definitions.

Figure S6 – The shaded region for the model 95% CI is not visible.

Reviewer #3 (Remarks to the Author):

Overall, I think this is a very nice research paper. The paper is well written, the study is described in sufficient detail, and the results are new and informative. In particular, quantifying how successful the vaccination program was and the counterfactual effects of other policy/public health decisions in French Polynesia is important for reflecting on how any future similar epidemics could be handled. I have a handful of suggestions, which I list below:

I have two primary concerns about the study, which may be indirectly related. The first is regarding the narrowness of the confidence intervals. I would expect much greater uncertainty in the system, especially given the relatively small numbers of cases, hospitalisations, and deaths at any given time point. Clearly the data falls outside the CI most of the time. Do you think this has arisen because the parameters are too highly constrained? My second concern is the decision to run one very long MCMC chain rather than 3-4 shorter chains, which is a more common approach. The potential issue I see here is that running multiple chains from different initial conditions will provide useful information about how well the model estimates the parameters. Further, with multiple chains you can use metrics to quantify convergence.

Another suggestion, which is straightforward but not trivial, is to run a sensitivity analysis. This is a very complicated model and most of the parameters are set based on data from other studies. It would be helpful to know how uncertainty in the model is allocated across these parameters.

Figure suggestions:

It might be useful to add a new two panel figure to either the main text or supplemental with a timeline of the outbreak with indications for different waves, variants, vaccine rollouts and NPIs along with a plot of the age distribution of the population. This could almost replace the entire second paragraph of the introduction and would be nice for readers to quickly look back at as they go through the results. This would further free up introduction text to incorporate some more

synthesized background information. Since the age distribution is so important for the model structure and results, it would also be nice to see it depicted.

Table 1 might be easier to understand with bar graphs: the zero line could be the fitted 'no change' model with separate bars (with CI) for each counterfactual scenario. Bars above (below) the zero line would indicate more (fewer) cases/hospitalisations/hospital deaths.

Figure 2: Given how sensitive the model is to booster assumptions, authors may want to consider including two 'no boosters' scenarios.

Other msc suggestions:

- What is a fitted reporting factor? Does 1 = a perfect fit? Is 0.55 good?
- It looks like the model underestimates deaths in the second wave for all age groups, not just 60+, is that not correct (albeit the death numbers overall are incredibly low broken down by age).
- I like that the authors clearly state that the no lockdown scenario assumes patients were managed the same. How realistic is this assumption? Do the hospitals have the capacity to handle a wave with double the number of patients? If not, this seems important to state given the likelihood of cascading effects with an overwhelmed healthcare system (as indicated in the response to the delta wave). If not, this is an important and likely controversial result.
- I'm not sure what is meant by the model doesn't 'capture the flatness of the first wave of hospitalizations and hospital deaths'.
- In the immune status section, I'm confused by the sentence '...most infections in the Delta wave were among unvaccinated uninfected individuals', what does 'unvaccinated uninfected' mean here? Do you mean no prior infection?
- In general, the discussion is simply rehashing the results section rather than placing it within the greater context of the field. I think the space could be better used. There is one whole paragraph dedicated to comparison with one other similar study (which seems like too much emphasis), and otherwise, there is little contextualization within the region or COVID19/respiratory diseases field.
- There is placeholder text with the words '(author?)' on p 7 & 12
- This is not an actionable suggestion and more of a note in case it's useful for future work. This type of model is much easier to specify in Python as there is a function to stratify the model. Therefore, you can write the base model, and then use single lines of code to stratify by specified groups (like age, vaccine status, etc.). There are even software developed for modular epidemiological models like this one (e.g., <https://github.com/monash-emu/AuTuMN>).

Jamie Caldwell

Response to reviewers

Summary of main changes

1. Added sensitivity analysis for vaccination, booster and NPI impact for parameters whose values are uncertain.
2. Ran multiple MCMC chains and computed convergence diagnostics to verify convergence of the MCMC.
3. Adjusted contact matrix for France by relative population density by age of French Polynesia to account for differences in demography.
4. Fitted overdispersion parameters for observation processes for cases, hospitalisations, and deaths, to avoid artificially constraining model fit, and added posterior predictive intervals to model fit plots to show uncertainty in model predictions accounting for observation uncertainty.
5. Fitted relative risk of hospitalisation for Delta variant compared to wild-type variant and time-varying probability of death given hospitalisation (with different values in each wave) to improve fit to hospitalisation and death data and accuracy of estimates of hospitalisations and deaths averted.
6. Provided further justification of modelling assumptions made and discussion of limitations of analysis.
7. Provided further explanation in the main text of differences to existing COVID-19 modelling studies and novelty.

All page numbers for the main text in the following responses refer to the version with tracked changes, 'main_revised_track_changes.pdf'.

Responses to reviewers comments

Reviewer #1 (Remarks to the Author):

Due to the small scale of the study and the topic, which has been explored at length for numerous countries and contexts (being of great relevance in 2021 and 2022 for COVID control), the results are not in themselves noteworthy in novelty or applicable in terms of policymaking. Global studies which have explored the questions posed by this paper have already been conducted (e.g. Watson et al 2022;

[https://www.thelancet.com/journals/laninf/article/PIIS1473-3099\(22\)00320-6/fulltext](https://www.thelancet.com/journals/laninf/article/PIIS1473-3099(22)00320-6/fulltext), Moore et al 2021; [https://www.thelancet.com/journals/laninf/article/PIIS1473-3099\(21\)00143-2/fulltext](https://www.thelancet.com/journals/laninf/article/PIIS1473-3099(21)00143-2/fulltext))

We agree that both of these studies address superficially similar questions to the ones we explore in the paper, and we referenced both of them in the introduction of the paper. However, there are a number of key differences between our study and these studies. First, both of these studies consider only the first year of vaccine rollout up to

late 2021, excluding the Omicron wave for many countries, and therefore don't estimate the counterfactual impact of booster vaccinations on hospitalisations and deaths during the first Omicron wave for most countries. In contrast, we consider epidemic dynamics up to May 2022, and thus include the first Omicron wave in French Polynesia, and estimate the impact of booster vaccinations on severe outcomes during the Omicron wave. As we point out in the introduction, neither of the models in these studies incorporate different variants explicitly, and thus they are not able to explicitly account for differing cross-immunity between variants and variant severity, or the array of different levels of immunity that exist from different infection and vaccination histories. Second, both studies use estimated excess death data as the main data source when fitting their model, a necessary approximation when considering multiple settings with limited surveillance. In contrast, we use multiple detailed data streams, including hospitalisations, hospital deaths and seroprevalence, and provide a robust inference algorithm for linking these data sources. Studies like Watson et al and Moore et al are valuable for broad-level global comparisons, but our study therefore provides a more comprehensive approach to estimating vaccine impact for countries for which multiple data sources are available. We would argue that the framework and inference approach we provide to integrate multiple data sources to estimate vaccination impact – while accounting for multiple variants and complex infection and vaccination histories – is therefore a novel contribution both in terms of methodological development and providing previously undocumented insights from an isolated population.

Although the methodology is overall sound as separate components (except for potential notation issues identified below), the use of the SEIR (or similar frameworks) modelling to address this questions has been done repeatedly and is not novel, nor does it contribute significantly to our understanding of COVID-19 control or vaccination at this time. Examples include Aruffo et al. 2022; <https://bmcpublichealth.biomedcentral.com/articles/10.1186/s12889-022-13597-9> and Li et al. 2021; <https://www.frontiersin.org/articles/10.3389/frai.2021.648579/full>). Therefore I paid more attention to other aspects of the paper including the parallel flow, but found it challenging to accept the results due to the high complexity added for some components, and gross simplification of others.

The use of simple compartmental frameworks in epidemiology is indeed not new, and such approaches can be traced back well over a hundred years. However, the use of data synthesis and statistical inference that accounts for multi-strain epidemic dynamics as in this paper is an area where there remains considerable methodological development. Likewise, heterogeneity in COVID burden globally and the relative impact of control measures continues to stimulate active debate (1), particularly against a complex background of vaccine and infection-induced immunity.

It should also be noted that many countries experienced multiple waves of the same or differing variants (e.g. Screen et al 2022; https://wwwnc.cdc.gov/eid/article/28/13/22-0228_article) and have created models to explore this (e.g. LaJoie et al 2022; <https://www.nature.com/articles/s41598-022-24967-z>)

We have now placed these studies in context in the Introduction, adding the suggested reference and others on p2:

“Many countries have experienced multiple waves from the same or different variants (2–4). ... While frameworks for modelling multiple variants have been developed (5–9), most are country specific and not straightforwardly generalisable to other settings, or do not provide robust and flexible inference methodology for fitting to multiple data streams. Here we develop a framework that explicitly addresses these issues and apply it to COVID-19 epidemic waves in French Polynesia.”

We have also added the following paragraph discussing the LaJoie 2022 paper and other multistrain modelling frameworks in the *Discussion* on pp. 6-7:

“Many existing modelling frameworks for estimating COVID-19 vaccination and NPI impact (10–14) do not explicitly account for the complicated COVID-19 immune landscape that now exists, with different levels of protection against different outcomes from different variants due to varying vaccination and infection histories and variant properties (transmissibility, immune escape, and severity). Those that do (5,7–9,15) tend to be country specific; reliant on detailed infection prevalence, hospitalisation and/or mobility data; and not readily transferable to estimate vaccination and NPI impact in settings without such data. We have developed an age-structured multi-strain SARS-CoV-2 transmission model that addresses this gap, and used it to estimate the impact of vaccination and non-pharmaceutical interventions on incidence of cases and severe outcomes in the first three waves of COVID-19 in French Polynesia. While our approach is similar to the ‘stacked’ SIR-type multistrain transmission model of different levels of immunity from vaccination and infection developed by LaJoie et al. (6), we also stratify by age and thus are able to model age-dependent mixing and infer variation in immunity over time by age group, and we use a more robust and flexible framework for performing parameter inference across multiple data sources rather than just reported cases.”

The complexity of the model also makes interpretation and disentangling the impact of each component challenging, also unfortunately leaving many uncertainties due to the numerous assumptions and simplifications built one on top of the other.

We acknowledge that the model is relatively complex but would argue that some degree of model complexity is inevitable now when analysing COVID-19 dynamics to infer quantities such as the breakdown of immunity by age over time, due to the large number of possible combinations of infection and vaccination that individuals can have had and the impact these have had on their level of immunity and risk of different subsequent outcomes (reinfection, severe disease etc.).

At the same time, as for all models, it is necessary to make some simplifying assumptions to have a model that captures the essential elements of the process but remains computationally tractable. Where we have made simplifying assumptions we have noted them and either justified them with references or discussed them in the limitations (on pp. 8-9 in the main text and Section 3.1 on p17 in the Supplementary Information). We have also now quantified at least some of the uncertainty in our estimates of cases, hospitalisations, and deaths averted through vaccinations, boosters, and lockdowns due to uncertainty in certain parameters for which we had previously assumed fixed values (including the duration of natural immunity, cross-immunities to infection with Delta and Omicron from previous infection, booster waning rate, and vaccine effectiveness) by conducting a sensitivity analysis. We have rerun the MCMC and counterfactual simulations for 'pessimistic' and 'optimistic' assumptions (with respect to vaccination and booster impact) about these parameters to obtain lower and upper bounds for our estimates. Details and results of the sensitivity analysis can be found in Sections 1.1 and 2.4 on pp. 1-2 and pp. 14-15 in the Supplementary Information. Across the uncertainty in all of these parameters, the variation in the impact estimates is not insubstantial – the estimated hospitalisations and hospital deaths averted through vaccination vary from 2520 to 3630 and 794 to 1197 respectively from the pessimistic to optimistic parameter assumptions and through boosters from 3 to 163 and 0 to 9. Importantly, however, this variation does not affect our qualitative findings, that the 1st and 2nd doses had a much larger impact in terms of hospitalisations and deaths averted than the booster doses.

For example, the assumption that Omicron and all sublineages are a single variant and that susceptibility to the same variant infection has a mean of 6 years when data shows the relationship is incredibly complex across and within the same variants with estimates of much smaller time intervals of reinfection existing (COVID-19 Forecasting Team 2023; [https://www.thelancet.com/journals/lancet/article/PIIS0140-6736\(22\)02465-5/fulltext](https://www.thelancet.com/journals/lancet/article/PIIS0140-6736(22)02465-5/fulltext)). Thus I would expect this to be incorporated in terms of the uncertainty as a different distribution could cause the results to be very different in understanding the underlying immunity of the population.

The reviewer is correct that we model Omicron BA.1 and BA.2 as a single variant and that this is a simplifying assumption. We make this choice because:

- Modelling Omicron BA.1 and BA.2 as separate variants would require a significant increase in the complexity of the model structure, which would come with several strong assumptions.
- Given the number of hospitalisations and deaths that occurred during the BA.1/BA.2 wave and the lower severity of BA.1 and BA.2 than the Delta variant, modelling BA.1 and BA.2 as separate variants would have a limited impact on the estimates of total hospitalisations and deaths averted through vaccinations and booster doses, which is the primary aim of the study. So the increase in model complexity required to model BA.1 and BA.2 separately would yield limited additional insight in terms of estimating the impact of the vaccination and booster programmes.

We note that the mean duration of natural immunity of 6 years is for reinfection with the same variant, and corresponds to complete loss of immunity, i.e. once this immunity wanes individuals return to being as susceptible to severe outcomes upon reinfection with the same variant as they were prior to being infected with that variant (as described under *Waning immunity* on p14). This estimate is based on a systematic review that estimated the average risk reduction against reinfection was 90.4% up to 10 months following infection (16). Although this estimate does not necessarily apply to reinfection with different Omicron sub-variants, we note that most of the studies reviewed by the COVID-19 Forecasting Team (17) considered reinfection with different variants (either among pre-Omicron variants or with Omicron BA.1 following infection with pre-Omicron variants) and estimates of protection against reinfection with an Omicron subvariant following infection with an Omicron subvariant ranged from 70-94%, while those for protection against reinfection with an Omicron variant following infection with a pre-Omicron variant were much lower, with a pooled estimate of 45%. There is also evidence that infection with an Omicron subvariant elicits strong neutralising antibody responses against infection with another Omicron subvariant (Supplementary Figure 2 (18)). Thus modelling pre-Omicron variants and Omicron variants separately would appear to be more important than modelling BA.1 and BA.2 separately for capturing the transmission dynamics sufficiently well to estimate vaccination and booster impact. We have edited the paragraph in the limitations on p17 in the Supplementary Information where the impact of modelling Omicron BA.1 and BA.2 as a single variant was discussed to reflect these points and include the COVID-19 Forecasting Team paper:

“Another potential source of underestimation of the impact of the booster programme is that we treat the Omicron BA.1 and BA.2 sublineages as the same, despite evidence that BA.2 is more transmissible than BA.1 (19–22) and can reinfect individuals previously infected with BA.1 (17), and may therefore cause higher infection rates in the

absence of protection from boosters. To some extent the difference in transmissibility will be absorbed in the fitting of the transmission rate for the third wave, but the resulting averaged transmission rate will give a lower estimate for the number of cases, hospitalisations and deaths averted in the latter half of the third wave when BA.2 was dominant and booster coverage was higher. Despite the potential for reinfection with BA.2 following BA.1 infection, estimates of the rate of such reinfection are reasonably low (6-30% (17)) and there is evidence that infection with BA.1 elicits strong neutralising antibody responses against infection with BA.2 (Supplementary Figure 2 (18)). Vaccine effectiveness and severity estimates for BA.2 are also similar to those for BA.1 (23–27). Therefore, given the lower severity of the Omicron variants, the absolute error in numbers of hospitalisations and deaths averted is likely to be small.“

In addition, given uncertainty in the mean duration of immunity to reinfection with the same variant, we have now included a shorter duration of natural immunity of 3 years in our sensitivity analysis (described in Section 1.1 on p1 in the Supplementary Information). As can be seen from the sensitivity analysis results in Section 2.4 on pp. 14-15 in the Supplementary Information, this does not qualitatively change our findings in terms of the relative impact of vaccinations and boosters.

The authors also state they used social contact data from France for the island states of French Polynesia, when the social dynamics are very different. The use of movement data or other social contact estimates from literature would be more appropriate here. The dynamics of spread would also be expected to differ substantially. They do state the limitation in the fact that these are islands - I think this cannot be ignored as movement restrictions with natural barriers in place would reduce spread drastically under lockdown conditions. This aspect of the study actually merits investigation and presents a more unique angle in comparison to the numerous homogeneous landscape studies which are available.

Unfortunately there is no mobility data or social contact data available for French Polynesia so we are forced to use contact data from another country. We have now stated this in the limitations in the Discussion on p8. Although French Polynesia has geographical and cultural specificities, the structure of the society (particularly in households, schools and workplaces which are the main parameters used to generate contact matrix) is more similar to France than it is to most other countries. Moreover, during the COVID-19 pandemic, measures implemented in French Polynesia to reduce the transmission of SARS-CoV-2 were based on those implemented in France. This assumption is something we have discussed extensively within the project team – many of whom live and work in French Polynesia, and were heavily involved in the local response – and think that among existing contact data (28) that from France is the most appropriate for French Polynesia.

We agree that the fact that the population is distributed on several islands limits movements compared to non-island countries. However, the majority (75%) of the population resides on Tahiti and Moorea and most inhabited islands had frequent air connections with Tahiti during the pandemic, except during lockdowns. Following discussions within the team, we believe the simplifying assumption of age-dependent-only mixing during non-lockdown periods is therefore reasonable. Whilst the suspension of inter-island flights during lockdowns would clearly have had an impact on mixing of the population, this will to some extent have been accounted for in the model in terms of the estimated values of the transmission rate parameters during the lockdowns in the first and second waves. We have mentioned this in the limitations on p9:

“The suspension of inter-island flights during lockdowns clearly would have had an impact on mixing of the population, but this is to some extent captured in the fitted values of the transmission rate parameters during the first and second epidemic waves.”

As we mention in the limitations on p9, in an ideal scenario we would have either focused the analysis on the Windward Islands (which include Tahiti and Moorea) or modelled transmission on the different archipelagos in a metapopulation model, but unfortunately even with the detailed data available, we did not have sufficient geolocation detail for cases to justify this more complex modelling approach:

“Ideally we would model transmission on the different archipelagos with a metapopulation model with the impact of lockdowns on inter-island movement informed by mobility data, or focus the analysis on the Windward Islands, which include Tahiti and Moorea, as these were most affected by the epidemic, but no mobility data is available for French Polynesia and we do not have sufficient geolocation detail for cases.”

The impact of interventions also has received much attention as aforementioned. Disentangling the impact of lockdown in itself is a highly complex modelling exercise with considerations on compliance by location, heterogeneity in impact across age, sources of infection still occurring within the community/home/workplaces etc. The addition of vaccination via a parallel flow with different vaccination pathways (i.e. variant 1 to 2, variant 2 to 1) ignores the complexity of interventions, and instead adds substantial detail in infection histories which is difficult to separate from intervention effectiveness with their model framework. An agent based model may be more appropriate here if such detail is to be modelled, which allows full exploration of all parameter space, acknowledgment of the complexity of human behaviour through simulation - this would be reflected in the wider confidence intervals in the results which

should exist considering both the complexity and the amount of parameter/distribution uncertainty which should exist in many of the parameters.

We believe that the added detail of modelling alternative infection and vaccination histories is in fact necessary to generate accurate estimates of vaccination and booster impact while accounting for heterogeneity in levels of immunity between age groups and individuals with different levels of previous infection and vaccination. In our opinion, there is sufficient data available from infection/reinfection, vaccine effectiveness, and variant severity studies with which to parameterise the different infection and vaccination pathways in the model and we have provided references to the relevant studies in Tables 4-6, and conducted a sensitivity analysis for the more uncertain parameters (Tables S1-S2).

While an agent based model might allow greater exploration of possible dynamics arising from complex human behaviour, it would likely be more assumption dependent than our model and would be much harder to fit to the data we have available, so would be liable to generate less robust conclusions. For example, many of the agent based models used to inform COVID policy have required more detailed GPS mobility or radio frequency identification contact data to parametrise.

The credible intervals around the results in the main text indicate the uncertainty in our estimates based on the data and model with our 'central' assumptions for fixed parameter values, i.e. do not include uncertainty in the fixed parameter values included in the sensitivity analysis. The much greater variation in the impact estimates from uncertainty in the fixed parameters is shown in Figs S8 and S9 and discussed in Section 2.4 in the Supplementary Information. We also note that the credible intervals around our results in the main text represent the uncertainty in the expected values of the different outcomes from our model (i.e. do not account for uncertainty in the processes via which cases, hospitalisations and deaths are reported). This was noted in the legend of Fig. 2 (p3) and we have added posterior predictive intervals to the plots of the model fit in Fig. 2 to show the uncertainty in the predicted values of each outcome from the model also accounting for uncertainty in the reporting process. Please also see our response below to Reviewer 3's comment about the narrowness of the credible intervals.

Whilst there is sufficient detail to repeat the exercise, the numerous assumptions would make the results difficult to stand by, especially for other contexts with larger populations and complex mixing, with longer periods of overlapping variants outbreaks or changing compliances to NPIs.

We hope that the changes detailed above, in particular the sensitivity analysis, the more detailed discussion of the limitations of the study, and the further explanations and justifications provided in the manuscript, have gone some way to assuage the reviewer's concerns. We also note that a similar model and framework based on the same underlying R simulation packages has been used for exactly the purpose the reviewer describes – modelling the epidemic in England with multiple overlapping waves caused by different variants to estimate vaccination impact and variant severity and transmissibility (15,29).

Minor issues:

A few citations are missing e.g. page 7, page 12

Thank you. This has now been corrected.

Might be wrong in page 14, Last two formulas on page 14, is it $(1 - e^{-\dots})$... rather than $e^{-\dots}$?

You are correct – thank you for spotting this. It's now been corrected.

For the population $J = 3, 4$, the author assumed them to be waning in the immunity and removed from R population but did not add them back in the S population as the formula accepts only $J = 1, 2$ from R population - is there a notation issue?

Yes, this was a typo in the equation for S, where the sum over nRS_{ijk} should have been from $j = 1$ to 4, and has now been corrected. Thanks for spotting this.

Confusion on reasoning for combining the variant 1 and 2 into one stratum (henceforth variant 1) when variant 3 is introduced (which is henceforth denoted as variant 2)?

We combine the information from the 1st and 2nd variant into the stratum for the first variant when a new variant is introduced to avoid the dimensionality of the model growing rapidly as the number of modelled variants increases and the model becoming too slow to fit. We have now explained this by adding the following sentence at the end of the description of the model in the *Model* section on p13:

“This simplification of the multistrain dynamics is to prevent the dimensionality of the model exploding as the number of variants and possible infection and vaccination histories increases, which would make the model prohibitively slow to fit.”

Reviewer #2 (Remarks to the Author):

Chapman et al. use an age-structured multi-strain COVID-19 transmission modeling framework to account for the impact of changes in underlying immunity on effectiveness of interventions, including vaccination and social distancing via lockdown. This is an important contribution to mathematical modeling of COVID-19 and the application to French Polynesia offers insights into the role vaccine-induced and exposure-based immunity have played. The manuscript is well written with detailed description of the model and data used to inform it. Suggestions below reflect model assumptions and findings that could warrant additional justification/comment.

Introduction:

- It would be helpful to have the seroprevalence study results (briefly) mentioned in the second paragraph to reflect seroprevalence within the timeline of the variants/waves/interventions.

We have now mentioned the seroprevalence study results in the second paragraph of the *Introduction* on p2 to place them within the context of the different waves/variants/interventions.

Results:

- Paragraph 2: Removal of lockdowns in the first and second waves was associated with overall reduction in hospitalizations and hospital deaths. It is understood that this means across the entire period under study (768 and 107). However, the authors then suggest that the same scenario led to reductions in wave 2 that were greater than the reductions overall (1430 vs 768 and 230 vs 107). Please revisit the sentences to reword if needed.

The reviewer is correct that the reductions are over the entire study period. The reductions in the second wave are counterbalanced by increases in the first wave. We have reworded these sentences as follows to clarify this (note that the estimates have changed due to improvements in the model described in our other responses):

“We estimated that removing the lockdowns in both the first and second waves would have had a non-linear effect on dynamics, and – assuming everything else had remained the same – would have led to fewer hospitalisations and hospital deaths over the study period from July 2020 to May 2022 (45 (95% CI -42–134) and 193 (95% CI 158–231) fewer, respectively) but a slightly higher number of symptomatic cases (1800 (95% CI 1300–2200) more). ... The non-linear effect on overall incidence is due to the first wave of infections being much larger (with 27,600 (95% CI 25,700–29,900) more symptomatic cases, 860 (95% CI 760–990) more hospitalisations, and 105 (95% CI 85–129) more deaths), resulting in greater build up of immunity in the population prior to

the introduction of the more severe Delta variant, and therefore a much smaller second wave of cases, hospitalisations and deaths (with 30,300 (95% CI 27,800–33,900) fewer cases, 940 (860–1040) fewer hospitalisations, and 299 (95% CI 263–338) fewer deaths).”

Methods:

- Thank you for the clear variant-specific vaccination data in Table 5. It appears that vaccination parameters on effectiveness and waning were assumed to not be age-specific. Is this decision based on literature or a simplifying assumption? Could the authors please support the decision with references if the former or note the decision in the limitations if the latter. (Note evidence on potential age differences in booster effectiveness -

<https://www.ecdc.europa.eu/en/publications-data/interim-analysis-covid-19-vaccine-effectiveness-against-severe-acute-respiratory>)

You are correct that the vaccine effectiveness and waning parameters are assumed to be the same across all age groups. This is a simplifying assumption. We have reworded the last sentence of the *Vaccination* section in the *Methods* on p14 as follows to emphasise this:

“We **make the simplifying assumption** that vaccine effectiveness is the same across all age groups.”

and we state in the *Waning immunity* subsection of the *Methods* on p14 that:

“We assume that the waning rates of natural and vaccine-induced immunity are the same for all age groups and virus variants.”

We did already discuss this limitation in the *Limitations* section of the Supplementary Information, where we wrote:

“We assume initial vaccine effectiveness and rates of waning of immunity are the same for all ages. However, there is evidence that vaccine effectiveness against symptomatic infection is lower and wanes more quickly in older age groups (≥ 65 years) than in younger age groups, at least for the Delta variant (30), and of potential age differences in booster effectiveness against Omicron variants (ECDC 2023), which may introduce some bias into our estimates of vaccination and booster impact. We also assume waning rates are the same for different variants and infection outcomes, but data suggests waning is faster against the Omicron BA.1 variant than the Delta variant (31) and that protection against severe outcomes wanes more slowly than that against

infection (30). Further work is needed to determine the extent to which these differences affect vaccine impact estimates.”

However, we have now moved this to the discussion of the limitations in the *Discussion* section on p9 of the main text, and added the reference to the document you cite.

- Assumptions around contact patterns, care-seeking behavior, and ‘effectiveness’ of hospital care (probability of death among hospitalized cases) are that French Polynesia is similar to France. It is understood that French Polynesia is a French territory. However, could data from other settings (perhaps island settings) be more representative of the situation in French Polynesia? Or are there any thoughts on how sensitive the model is to these assumptions that France is representative of French Polynesia? Please consider including a discussion of this.

We thank the reviewer for raising this important issue. Although French Polynesia has geographical and cultural specificities, its population age structure and the structure of its society (particularly in households, schools and workplaces, which are the main parameters used to generate the contact matrices) are more similar to France than other island countries. It is also more similar in wealth to France (it is classified as a high-income country, whereas other Pacific islands, such as Fiji and Tonga are classified as middle-income countries). During the pandemic, measures implemented in French Polynesia to reduce transmission of SARS-CoV-2 were based on those implemented in France. Consequently, we believe that contact data (28) and data on care-seeking behaviour and effectiveness of hospital care from France offers the best proxy for dynamics in French Polynesia.

To account for the impact of differences in the population age structure of France and French Polynesia on age contact patterns in French Polynesia, we now apply a correction to the contact matrix from France where we scale the number of contacts for individuals in each age group by the relative population proportions of the contact age groups in the two countries, following Prem et al (32) and Arregui et al (33). This corrects the numbers of contacts in the contact matrix for France by the density of available contacts in each age group in French Polynesia. Please see the *Force of infection* subsection of the *Methods* on p14 for full details.

We allow for some differences in care-seeking behaviour between France and French Polynesia by fitting the maximum (over age groups) probability of hospitalisation given symptomatic infection rather than using a fixed value for France from the literature. (It is only the relative risks of hospitalisation between age groups for which we use the literature values.) We obtain a value of 0.23 for this probability compared to the estimate

of 0.50 from the literature (combining (34,35)). We also now fit the probability of death given hospitalisation in the different epidemic waves rather than assuming that it is the same as for France and constant over time, to account for differences in the effectiveness of hospital care between the two countries and over time within French Polynesia. We find considerable variation in the probability of death among hospitalised cases, increasing from 0.22 in the first wave up to a peak of 0.75 at the height of the Delta wave and down to 0.09 in the Omicron BA.1/BA.2 wave (cf. the literature estimate for France from 2020 of 0.32 (34)).

The above changes have improved the fit of the model to the hospitalisation and death data for the Delta and Omicron BA.1/BA.2 waves, and have led to some quantitative changes in the estimated impact of vaccination and the booster programme, but not changed the findings qualitatively.

- It is noted that the authors fit a time-varying transmission parameter to account for impact of lockdown and use probability of hospitalization for care-seeking (based on data from France). However, given the focus of NPIs and vaccination in the paper, please consider explicitly adding a section of text to the Methods to account for how behavior parameters (care-seeking, vaccine/booster-seeking, and lockdown adherence) are addressed in the model, particularly across age and vaccination strata, and natural immunity compartments.

As suggested, we have added the following section to the *Methods* on pp. 15-16:

“Behaviour

The impact of lockdowns on transmission is described in the model through a time-varying transmission rate, with changepoints corresponding to major changes in restrictions in French Polynesia (Figure 1). We make the simplifying assumption that adherence to these restrictions is the same across all age groups and vaccination strata, and regardless of infection history. Variation in care-seeking behaviour with age is modelled through an age-dependent probability of hospitalisation given symptomatic infection, where the relative risks of hospitalisation between age groups are based on data from France (34) and we estimate the maximum probability of hospitalisation across all age groups to account for differences in care-seeking and access to care between France and French Polynesia. The probability of hospitalisation varies across vaccination strata in the model due to the different levels of protection against severe disease with different levels of vaccination described above (see *Vaccination*), but we do not model any variation in care-seeking behaviour with vaccination status beyond this. Vaccine and booster uptake by age and vaccination status in the model are determined by the data on the numbers of each dose received by age over time (Figure 4), assuming that all individuals within each age-and-vaccination stratum have an equal

chance of being vaccinated (i.e. previous infection does not affect vaccine/booster-seeking) and can only receive successive vaccine doses (e.g. must have had the second dose to receive a booster).”

Please also see our response to your previous comment.

Discussion:

- The authors’ work suggests that lockdowns rendered the situation worse for waves 2 and 3. This is understood in the context of immunity. However, could they better contextualize this in terms of country-level resources for response? More specifically the authors assumed sufficient and uniform capacity for care despite increased case counts. Please consider commenting on the number of available hospital beds and whether the increase in case counts during the first wave would have exceeded this capacity. I see a note in the Supplement, but it seems like an important point (often contentious, multifaceted considerations that go into lockdown decisions) that may warrant attention in the Main Text, particularly as decision-makers may look to this evidence for response decisions in future/non-COVID outbreaks.

Based on our estimates, removing the first lockdown would have brought the peak incidence of hospitalisations in the first wave close to but slightly lower than that that occurred in the second wave (see top middle panel in Fig. S9 on p15 of the Supplementary Information), when hospital capacity was reached. Since it was possible to make 248 general beds available for hospitalised cases at CHPF (Centre Hospitalier de la Polynésie française), French Polynesia’s main hospital where most COVID-19 patients were treated, and ICU bed capacity had already been upgraded to 36 in August 2020 (with the army placed on standby to provide an additional 10 beds), and the number of hospitalised cases at CHPF peaked at 246 (48 in the ICU) in the second wave, hospital capacity would not necessarily have been exceeded in the first wave under the ‘no lockdown’ scenario. Nevertheless, it may not have been possible to maintain the same level of care under increased pressure on hospital resources. We have now highlighted this in the *Discussion* by adding the following text on p9:

“We assume that if lockdowns had been removed and hospitalisations had increased by nearly 75% during the first wave, hospital capacity would not have been exceeded and hospitalised patients would have received the same quality of care. Based on our estimates, the peak incidence of hospitalisations in the first wave would have been slightly lower than that that occurred during the second wave (Figure S9), when the number of general hospitalised and ICU COVID-19 cases at Centre Hospitalier de la Polynésie française (CHPF), the main hospital in French Polynesia where most COVID-19 patients were treated, peaked at 246 and 48 respectively. Since it was possible to make 248 general beds available for hospitalised cases at CHPF and ICU

bed capacity there had already been upgraded to 36 in August 2020 (with the army placed on standby to set up 10 more ICU beds if required), it is therefore not unreasonable to assume hospitals would have remained within capacity. Nevertheless, the increased pressure on hospital resources might have led to lower quality of care for hospitalised patients and hence poorer outcomes. The estimated reduction in overall hospitalisations and hospital deaths from removing lockdowns should thus be interpreted with caution.”

Supplement:

- Thank you for the extended discussion of Limitations. Could the authors also comment on how flexible the model is to address dynamics in other settings? Particularly, in settings where no seroprevalence data are available and/or less information is available on distributions of variants in the population. The model could provide valuable explanatory information for settings where lockdowns were less possible/enforced and vaccines (particularly boosters) have been less available, but these settings also have less data available. What minimum country-specific information is needed to generalize the model to other settings? Please consider discussing whether there should be any caution in using the model in a different context.

The model is certainly flexible enough that it could be used to address dynamics and provide estimates of vaccination and NPI impact in other settings, even those with less data and without seroprevalence and variant sequencing data. We have replaced the final sentence of the Discussion on p9 as follows to note this, to define the minimum data required to fit the model to other countries, and to say where caution is required in doing so:

“The framework is sufficiently flexible that it could be used to model COVID-19 dynamics and estimate vaccination and NPI impact for other countries, including those with less data available. As a minimum, COVID-19/excess death or hospitalisation data, case or seroprevalence data, vaccination and booster coverage data, dates of major changes in restrictions, and broad date ranges for the introduction of different variants would be required to fit the model, but in such a scenario all parameters except for the time-varying transmission rate, variant introduction dates, and symptomatic case reporting rate would need to be fixed to avoid parameter identifiability issues. In settings without seroprevalence data, case or regular testing data would be required to infer infection levels, or a fixed infection-hospitalisation/infection-fatality rate would have to be assumed. For countries with no variant sequencing data, date ranges for the introduction of different variants would have to be based on variant introduction dates for countries in that region or estimates of global emergence dates of new variants. Caution would be required to only apply the model to countries for which COVID-19 hospitalisation/death or excess death data was deemed to be reasonably complete to

avoid biased estimates of impact. Nonetheless, the framework could still provide valuable insight in settings with different vaccine and booster availability and NPI levels.”

- *In the Main Text, the authors note that “The model reproduces the overall patterns in the data, although it does not fully capture the flatness of the first wave of hospitalisations and hospital deaths, underestimates deaths among 60+ year-olds in the second wave, and overestimates hospitalisations in the third wave.” Do any of the limitations presented here (or in the Discussion) address these observed variances? If so, please comment in the text.*

We have improved the model fit by allowing greater flexibility in the prior for the outbreak start date, estimating the relative risk of hospitalisation for the Delta variant compared to the wild-type virus, and fitting the probability of death given hospitalisation for each wave, which has rendered the first and last of these observations obsolete. We have deleted those parts in the *Model fit* subsection of the *Results* on p2 and updated the relevant text in the discussion of the limitations (on p17 in the Supplementary Information) as follows:

“Although we fit the maximum probability of death given hospitalisation in each wave and thus account for the considerable variation in the risk of hospital death over time, we still underestimate the number of hospital deaths among individuals aged 60-69 years during the Delta wave (by approximately 20%). The increased risk of death in this age group in the Delta wave may be related to the main hospitals reaching capacity at its peak and it not being possible to provide the highest risk patients with the same level of care. We do not account for any such age-specific effect of hospital capacity in the model. Hospitals reaching capacity may also have led to increased death rates in the community (since some individuals with severe disease would not have been admitted). As the model does not account for this, we may have underestimated the transmission rate during this period and thus the counterfactual number of deaths without vaccination (as hospital capacity in the counterfactual scenario would have been reached more quickly, leading to a faster accumulation of deaths in the community).”

Figures:

Figure 3 on relative immune statuses over time is very helpful to see. Well done.

Thank you.

Figure 5 legend – There seem to be five vaccination strata: $k=1$ through $k=5$. Please check the subscript definitions.

Thank you for spotting this. Now corrected to “ $k \in \{1, 2, 3, 4, 5\}$ ”

Figure S6 – The shaded region for the model 95% CI is not visible

Thanks for noting this. We have increased the scale of the vertical axis and used a dashed line for the median, so that the 95% CI is now visible.

Reviewer #3 (Remarks to the Author):

Overall, I think this is a very nice research paper. The paper is well written, the study is described in sufficient detail, and the results are new and informative. In particular, quantifying how successful the vaccination program was and the counterfactual effects of other policy/public health decisions in French Polynesia is important for reflecting on how any future similar epidemics could be handled. I have a handful of suggestions, which I list below:

We thank the reviewer for their positive feedback.

I have two primary concerns about the study, which may be indirectly related. The first is regarding the narrowness of the confidence intervals. I would expect much greater uncertainty in the system, especially given the relatively small numbers of cases, hospitalisations, and deaths at any given time point. Clearly the data falls outside the CI most of the time. Do you think this has arisen because the parameters are too highly constrained? My second concern is the decision to run one very long MCMC chain rather than 3-4 shorter chains, which is a more common approach. The potential issue I see here is that running multiple chains from different initial conditions will provide useful information about how well the model estimates the parameters. Further, with multiple chains you can use metrics to quantify convergence.

We are grateful to the reviewer for raising these concerns. We do not think the narrowness of the credible intervals is due to the parameters being too highly constrained. The credible intervals shown on the plots of the model fits to the different data streams in Fig. 2 and Figs S4-S6 are 95% quantile intervals for the numbers of confirmed cases, hospitalisations and hospital deaths in the model, i.e. show the uncertainty in the expected values of the outcomes from the model and do not account for uncertainty in the observation processes (as we mentioned in the legends). Thus, their narrowness represents how highly constrained the expected (i.e. “true” unobserved underlying) numbers of cases, hospitalisations and deaths are for the given data and model. Nevertheless, to ensure that we do not constrain parameters to incorrect values by assuming values for the overdispersion in the negative binomial observation processes for the case, hospitalisation and death data streams, we now fit the overdispersion parameters for these distributions. We have also now added 95% posterior predictive intervals to these plots to show the uncertainty in model-predicted

values for the different outcomes including uncertainty from the observation processes, and described this in the figure legends. As you can see from the updated figures, the posterior predictive intervals cover 95% of the data points, indicating that the model is well calibrated.

We agree with the reviewer about the benefits of running multiple MCMC chains rather than a single chain, and have now run 4 chains to verify that the MCMC has converged and to calculate convergence diagnostics, namely the maximum value of the Gelman-Rubin statistic and minimum effective sample size across all the parameters. We have updated Fig. S1 on p4 in the Supplementary Information to show trace plots for all the chains and updated the text in Section 2.1 on p3 as follows:

“The trace plots for the fitted parameters in Figure S1 show successful convergence of the MCMC, as the chains, which were started from different points in the 17-dimensional parameter space, all converge to the same values. Similarly, the maximum Gelman-Rubin diagnostic across all the parameters for the combined chains (4000 iterations) of 1.02 (which is well below the recommended threshold of 1.1) and the minimum effective sample size of 518 indicate that the chains have converged.”

Another suggestion, which is straightforward but not trivial, is to run a sensitivity analysis. This is a very complicated model and most of the parameters are set based on data from other studies. It would be helpful to know how uncertainty in the model is allocated across these parameters.

We thank the reviewer for this suggestion. We have now conducted a sensitivity analysis to determine the sensitivity of the estimated numbers of cases, hospitalisations and deaths averted through lockdowns, vaccination, and boosters to variation in key uncertain parameters, including the duration of natural immunity, booster waning rate and vaccine effectiveness. Details and results of the sensitivity analysis can be found in Sections 1.1 and 2.4 on pp. 1-2 and pp. 14-15 in the Supplementary Information. Due to the need to rerun the MCMC and counterfactual simulations for each different set of fixed parameters considered, conducting sensitivity analyses for each uncertain parameter in turn or a multivariate analysis would be extremely computationally costly, so we have instead chosen the pragmatic approach of identifying lower and upper bounds for each of the uncertain parameters and rerunning the MCMC and counterfactual simulations with the sets of combined most “pessimistic” parameter assumptions and combined most “optimistic” assumptions with respect to vaccination and booster impact (our primary outcomes of interest). Whilst this does not permit identification of the sensitivity of the estimates to individual parameters, it does provide lower and upper bounds for the vaccination and booster impact given uncertainty in these parameters. We find that the estimated number of symptomatic cases averted

through vaccination is relatively constant across the different parameter values considered, but the estimated numbers of hospitalisations and deaths averted are somewhat sensitive to uncertainty in the parameters (the former varying from 2520 to 3630 and the latter from 794 to 1197 from the pessimistic to optimistic parameter assumptions). The estimated booster impact is also quite sensitive to uncertainty in the parameters (see Fig. S8 on p14 of the Supplementary Information).

Figure suggestions:

It might be useful to add a new two panel figure to either the main text or supplemental with a timeline of the outbreak with indications for different waves, variants, vaccine rollouts and NPIs along with a plot of the age distribution of the population. This could almost replace the entire second paragraph of the introduction and would be nice for readers to quickly look back at as they go through the results. This would further free up introduction text to incorporate some more synthesized background information. Since the age distribution is so important for the model structure and results, it would also be nice to see it depicted.

We have added the suggested figure as the new Fig. 1 on p2 in the main text. We have provided further contextualisation in the Introduction as described in our other responses (please see changes in tracked changes version).

Table 1 might be easier to understand with bar graphs: the zero line could be the fitted 'no change' model with separate bars (with CI) for each counterfactual scenario. Bars above (below) the zero line would indicate more (fewer) cases/hospitalisations/hospital deaths.

We agree that this information is more easily understood visually and have now added the suggested bar graphs as an extra panel in Fig. 3 on p4, to synthesize the information with the plots showing the numbers of cases, hospitalisations, and hospital deaths over time under the different counterfactual scenarios. We have moved the previous table to the Supplementary Information for completeness.

Figure 2: Given how sensitive the model is to booster assumptions, authors may want to consider including two 'no boosters' scenarios.

Thanks to the reviewer for the suggestion. Given we now have two alternative 'no booster' scenarios for the pessimistic and optimistic parameter assumptions and Fig. 3 would be too cluttered to be readable if we put all of these on the plots, we've included a separate figure (Fig. S9) on p15 in the Supplementary Information showing the numbers of cases, hospitalisations, and deaths over time for all counterfactual and parameter scenarios.

Other msc suggestions:

- *What is a fitted reporting factor? Does 1 = a perfect fit? Is 0.55 good?*

The “fitted reporting factor” is the estimated proportion of symptomatic cases that were reported based on fitting our model to the data. It is defined in Table 8 on p21 and denoted ϕ_{cases} . So, no, 1 does not indicate a perfect fit, and 0.47 (the new value due to changes to the model we have detailed in the rest of our responses) is a reasonable value based on a very rough, simplified calculation from our estimate of approximately 80% of the population (~224,000 people) having been infected by May 2022 (Table 1, p5) and roughly 50% of these having been symptomatic (~112,000 people) and 74,000 cases having been reported (giving a very rough reporting rate of 0.66). We have now rephrased the relevant sentence under *Model fit* on p2 to say:

“The estimated number of symptomatic cases over time corresponds closely to the numbers of confirmed cases during the three waves (Figure 1), with an estimated reporting rate of 0.47 (95% CI 0.46–0.49), i.e. 47% of symptomatic cases having been reported.”

- *It looks like the model underestimates deaths in the second wave for all age groups, not just 60+, is that not correct (albeit the death numbers overall are incredibly low broken down by age).*

The model was underestimating deaths in the second wave in all age groups, but only slightly in under-60s. (Part of the appearance of underestimation in the plots is due to the model outputs taking continuous values and the death counts being very low and highly dispersed as a result of reporting effects). As mentioned above, we have now fitted separate probabilities of death given hospitalisation for each wave, which has improved the model fit and significantly reduced underestimation of deaths in the second wave (see the new Fig. S4 on p7 in the Supplementary Information and Fig. 2 on p3 in the main text).

- *I like that the authors clearly state that the no lockdown scenario assumes patients were managed the same. How realistic is this assumption? Do the hospitals have the capacity to handle a wave with double the number of patients? If not, this seems important to state given the likelihood of cascading effects with an overwhelmed healthcare system (as indicated in the response to the delta wave). If not, this is an important and likely controversial result.*

We believe that this is not an unreasonable assumption, since the estimated peak incidence of hospitalisations under the 'no lockdown' scenario is slightly lower than that that occurred in the second wave, and hospital bed capacity had already been increased in the main hospitals in August 2020. Nevertheless, we have now added the following text to the *Discussion* on p9:

“Based on our estimates, the peak incidence of hospitalisations in the first wave would have been slightly lower than that that occurred during the second wave (Figure S9), when the number of general hospitalised and ICU COVID-19 cases at Centre Hospitalier de la Polynésie française (CHPF), the main hospital in French Polynesia where most COVID-19 patients were treated, peaked at 246 and 48 respectively. Since it was possible to make 248 general beds available for hospitalised cases at CHPF and ICU bed capacity there had already been upgraded to 36 in August 2020 (with the army placed on standby to set up 10 more ICU beds if required), it is therefore not unreasonable to assume hospitals would have remained within capacity. Nevertheless, the increased pressure on hospital resources might have led to lower quality of care for hospitalised patients and hence poorer outcomes. The estimated reduction in overall hospitalisations and hospital deaths from removing lockdowns should thus be interpreted with caution.”

Please also see our response to Reviewer 2's first comment on the *Discussion*.

• *I'm not sure what is meant by the model doesn't 'capture the flatness of the first wave of hospitalizations and hospital deaths'.*

We meant that the model trajectories for hospitalisations and hospital deaths were more peaked than the data during the first wave, but the model fit has now been improved by allowing more flexibility in the prior for the start date of the outbreak and this no longer applies, so has been deleted.

• *In the immune status section, I'm confused by the sentence '...most infections in the Delta wave were among unvaccinated uninfected individuals', what does 'unvaccinated uninfected' mean here? Do you mean no prior infection?*

Yes, we mean no prior infection, We've updated this to read “most infections in the Delta wave were among unvaccinated individuals without prior infection”.

• *In general, the discussion is simply rehashing the results section rather than placing it within the greater context of the field. I think the space could be better used. There is one whole paragraph dedicated to comparison with one other similar study (which*

seems like too much emphasis), and otherwise, there is little contextualization within the region or COVID19/respiratory diseases field.

We have substantially revised the Discussion to compare our study with other modelling analyses of vaccination and NPI impact; compare French Polynesia's pandemic trajectory with that of other Pacific Island Countries; and place our study within the broader context of COVID-19 modelling. Please see the highlighted changes on pp. 6-9 of the tracked changes version of the main text.

- *There is placeholder text with the words '(author?)' on p 7 & 12*

Thanks to the reviewer for noting this – it has now been corrected.

- *This is not an actionable suggestion and more of a note in case it's useful for future work. This type of model is much easier to specify in Python as there is a function to stratify the model. Therefore, you can write the base model, and then use single lines of code to stratify by specified groups (like age, vaccine status, etc.). There are even software developed for modular epidemiological models like this one (e.g., <https://github.com/monash-emu/AuTuMN>).*

Thanks very much to the reviewer for this note. We will bear it in mind for future work.

Jamie Caldwell

References

1. The Royal Society's programme on the impact of non-pharmaceutical interventions on Covid-19 transmission [Internet]. [cited 2023 Sep 5]. Available from: <https://royalsociety.org/topics-policy/projects/impact-non-pharmaceutical-interventions-on-covid-19-transmission/>
2. Smith-Sreen J, Miller B, Kabaghe AN, Kim E, Wadonda-Kabondo N, Frawley A, et al. Comparison of COVID-19 Pandemic Waves in 10 Countries in Southern Africa, 2020-2021. *Emerg Infect Dis.* 2022 Dec;28(13):S93–104.
3. Lin L, Zhao Y, Chen B, He D. Multiple COVID-19 Waves and Vaccination Effectiveness in the United States. *Int J Environ Res Public Health* [Internet]. 2022 Feb 17;19(4). Available from: <http://dx.doi.org/10.3390/ijerph19042282>
4. Ge Y, Zhang WB, Wu X, Ruktanonchai CW, Liu H, Wang J, et al. Untangling the changing impact of non-pharmaceutical interventions and vaccination on European COVID-19 trajectories. *Nat Commun.* 2022 Jun 3;13(1):3106.
5. Yang W, Shaman J. Development of a model-inference system for estimating

- epidemiological characteristics of SARS-CoV-2 variants of concern. *Nat Commun.* 2021 Sep 22;12(1):5573.
6. LaJoie Z, Usherwood T, Sampath S, Srivastava V. A COVID-19 model incorporating variants, vaccination, waning immunity, and population behavior. *Sci Rep.* 2022 Nov 27;12(1):20377.
 7. Keeling MJ, Dyson L, Tildesley MJ, Hill EM, Moore S. Comparison of the 2021 COVID-19 roadmap projections against public health data in England. *Nat Commun.* 2022 Aug 22;13(1):4924.
 8. Barnard RC, Davies NG, Centre for Mathematical Modelling of Infectious Diseases COVID-19 working group, Jit M, Edmunds WJ. Modelling the medium-term dynamics of SARS-CoV-2 transmission in England in the Omicron era. *Nat Commun.* 2022 Aug 19;13(1):4879.
 9. Lustig A, Vattiato G, Maclaren O, Watson LM, Datta S, Plank MJ. Modelling the impact of the Omicron BA.5 subvariant in New Zealand. *J R Soc Interface.* 2023 Feb;20(199):20220698.
 10. Watson OJ, Barnsley G, Toor J, Hogan AB, Winskill P, Ghani AC. Global impact of the first year of COVID-19 vaccination: a mathematical modelling study. *Lancet Infect Dis.* 2022 Sep;22(9):1293–302.
 11. Moore S, Hill EM, Dyson L, Tildesley MJ, Keeling MJ. Retrospectively modeling the effects of increased global vaccine sharing on the COVID-19 pandemic. *Nat Med.* 2022 Nov;28(11):2416–23.
 12. Marziano V, Guzzetta G, Mammone A, Riccardo F, Poletti P, Trentini F, et al. The effect of COVID-19 vaccination in Italy and perspectives for living with the virus. *Nat Commun.* 2021 Dec 14;12(1):7272.
 13. Yamana TK, Galanti M, Pei S, Di Fusco M, Angulo FJ, Moran MM, et al. The impact of COVID-19 vaccination in the US: Averted burden of SARS-COV-2-related cases, hospitalizations and deaths. *PLoS One.* 2023 Apr 25;18(4):e0275699.
 14. Shoukat A, Vilches TN, Moghadas SM, Sah P, Schneider EC, Shaff J, et al. Lives saved and hospitalizations averted by COVID-19 vaccination in New York City: a modeling study. *Lancet Reg Health Am.* 2022 Jan;5:100085.
 15. Perez-Guzman PN, Knock E, Imai N, Rawson T, Elmaci Y, Alcada J, et al. Epidemiological drivers of transmissibility and severity of SARS-CoV-2 in England. *Nat Commun.* 2023 Jul 17;14(1):4279.
 16. Kojima N, Shrestha NK, Klausner JD. A Systematic Review of the Protective Effect of Prior SARS-CoV-2 Infection on Repeat Infection. *Eval Health Prof [Internet].* 2021 Sep 30 [cited 2023 Sep 5]; Available from: <https://journals.sagepub.com/doi/full/10.1177/01632787211047932?cookieSet=1>
 17. COVID-19 Forecasting Team. Past SARS-CoV-2 infection protection against re-infection: a systematic review and meta-analysis. *Lancet.* 2023 Mar 11;401(10379):833–42.

18. Carr EJ, Wu MY, Gahir J, Harvey R, Townsley H, Bailey C, et al. Neutralising immunity to omicron sublineages BQ.1.1, XBB, and XBB.1.5 in healthy adults is boosted by bivalent BA.1-containing mRNA vaccination and previous Omicron infection. *Lancet Infect Dis*. 2023 Jul;23(7):781–4.
19. Lyngse FP, Kirkeby CT, Denwood M, Christiansen LE, Mølbak K, Møller CH, et al. Household transmission of SARS-CoV-2 Omicron variant of concern subvariants BA.1 and BA.2 in Denmark. *Nat Commun*. 2022 Sep 30;13(1):5760.
20. Yamasoba D, Kimura I, Nasser H, Morioka Y, Nao N, Ito J, et al. Virological characteristics of the SARS-CoV-2 Omicron BA.2 spike. *Cell*. 2022 Jun 9;185(12):2103–15.e19.
21. Ito K, Piantham C, Nishiura H. Estimating relative generation times and reproduction numbers of Omicron BA.1 and BA.2 with respect to Delta variant in Denmark. *Math Biosci Eng*. 2022 Jun 21;19(9):9005–17.
22. Lentini A, Pereira A, Winqvist O, Reinius B. Monitoring of the SARS-CoV-2 Omicron BA.1/BA.2 lineage transition in the Swedish population reveals increased viral RNA levels in BA.2 cases. *Med*. 2022 Sep 9;3(9):636–43.e4.
23. Wolter N, Jassat W, DATCOV-Gen author group, von Gottberg A, Cohen C. Clinical severity of omicron lineage BA.2 infection compared with BA.1 infection in South Africa. *Lancet*. 2022 Jul 9;400(10346):93–6.
24. Sievers C, Zacher B, Ullrich A, Huska M, Fuchs S, Buda S, et al. SARS-CoV-2 Omicron variants BA.1 and BA.2 both show similarly reduced disease severity of COVID-19 compared to Delta, Germany, 2021 to 2022. *Euro Surveill* [Internet]. 2022 Jun;27(22). Available from: <http://dx.doi.org/10.2807/1560-7917.ES.2022.27.22.2200396>
25. Lewnard JA, Hong VX, Patel MM, Kahn R, Lipsitch M, Tartof SY. Clinical outcomes associated with SARS-CoV-2 Omicron (B.1.1.529) variant and BA.1/BA.1.1 or BA.2 subvariant infection in Southern California. *Nat Med*. 2022 Sep;28(9):1933–43.
26. Kirsebom FCM, Andrews N, Stowe J, Toffa S, Sachdeva R, Gallagher E, et al. COVID-19 vaccine effectiveness against the omicron (BA.2) variant in England. *Lancet Infect Dis*. 2022 Jul;22(7):931–3.
27. [No title] [Internet]. [cited 2023 Sep 5]. Available from: https://assets.publishing.service.gov.uk/government/uploads/system/uploads/attachment_data/file/1063424/Tech-Briefing-39-25March2022_FINAL.pdf
28. Prem K, van Zandvoort K, Klepac P, Eggo RM, Davies NG, Centre for the Mathematical Modelling of Infectious Diseases COVID-19 Working Group, et al. Projecting contact matrices in 177 geographical regions: An update and comparison with empirical data for the COVID-19 era. *PLoS Comput Biol*. 2021 Jul;17(7):e1009098.
29. Imai N, Rawson T, Knock ES, Sonabend R, Elmaci Y, Perez-Guzman PN, et al. Quantifying the effect of delaying the second COVID-19 vaccine dose in England: a mathematical modelling study. *Lancet Public Health*. 2023 Mar;8(3):e174–83.
30. Andrews N, Tessier E, Stowe J, Gower C, Kirsebom F, Simmons R, et al. Duration of Protection against Mild and Severe Disease by Covid-19 Vaccines. *N Engl J Med*. 2022 Jan

27;386(4):340–50.

31. Stowe J, Andrews N, Kirsebom F, Ramsay M, Bernal JL. Effectiveness of COVID-19 vaccines against Omicron and Delta hospitalisation, a test negative case-control study. *Nat Commun.* 2022 Sep 30;13(1):5736.
32. Prem K, Cook AR, Jit M. Projecting social contact matrices in 152 countries using contact surveys and demographic data. *PLoS Comput Biol.* 2017 Sep;13(9):e1005697.
33. Arregui S, Aleta A, Sanz J, Moreno Y. Projecting social contact matrices to different demographic structures. *PLoS Comput Biol.* 2018 Dec;14(12):e1006638.
34. Salje H, Tran Kiem C, Lefrancq N, Courtejoie N, Bosetti P, Paireau J, et al. Estimating the burden of SARS-CoV-2 in France. *Science.* 2020 Jul 10;369(6500):208–11.
35. Knock ES, Whittles LK, Lees JA, Perez-Guzman PN, Verity R, FitzJohn RG, et al. Key epidemiological drivers and impact of interventions in the 2020 SARS-CoV-2 epidemic in England. *Sci Transl Med [Internet].* 2021 Jul 14;13(602). Available from: <http://dx.doi.org/10.1126/scitranslmed.abg4262>

REVIEWERS' COMMENTS

Reviewer #1 (Remarks to the Author):

I thank the author for their detailed and thoughtful responses to my queries. I will elaborate on my key questions below.

Response to the question "Due to the small scale of the study and the topic, which has been explored at length for"

Another reviewer also discussed the model's flexibility in other settings with limited data. Did the author also consider the scenario where the data stream quality is unevenly distributed (i.e., where one source has more significant details and size than other sources)? Would this introduce bias in the model output? Also, I understand the author used hospitalisation death data to prevent overestimating the number of deaths; however, due to the limited size of the data, the fitting may be relatively poor quality, especially when it is age-stratified. Could I understand more on how this could be addressed?

Response to the question "It should also be noted that many countries experienced multiple waves of the same or" and "The impact of interventions also has received much attention as aforementioned. ..."

I would think the model may be hard to be generalised to other countries e.g. French Polynesia has a higher tourist-per-resident ratio than most other countries. The lockdown effect may have a more significant impact and be easier to enforce in this specific country than in other cities with high population densities.

Response to the question "Whilst there is sufficient detail to repeat the exercise, the numerous assumptions would"

This may have been extensively discussed, but to add to it - I think the results from this paper, where the counterfactual example of no lockdown results in lower cases, will be hard to replicate in different countries. Lockdowns in principle can be very effective but their impacts have been heterogeneous – they differ in type, compliance and duration. The relative remoteness and disconnectedness of Polynesia, as well as its population size, make it a relatively unique setting. I don't think this can be accounted for in the estimated values of the transmission rate parameters.

Overall comments:

It is hard to be convinced French Polynesia is a good microcosm for other countries. The model might be too complex to be replicated in other countries, and many assumptions are unrealistic in big cities. Nevertheless, this paper has produced many results and investigations on the effect of booster vaccination, cross-variant immunity and differing variant waves, which whilst they are key to important decision making, seem very tailored to French Polynesia. I would struggle to apply these methods to many of the islands in the Southeast Asia/South Pacific context.

Reviewer #2 (Remarks to the Author):

The authors put in notable effort to reflect all feedback from the reviewers. It is commendable.

- The results reflect novel consideration of the role of immune status on intensity of waves. While the authors have made several simplifying assumptions, the modeling framework is very well documented now and the Discussion reflects on its flexibility for other settings, as well as the Limitations that might be overcome with additional data, for instance.
- The authors' clarification of language around the model fit and the addition of more discussion about how the model compares to other frameworks offers stronger evidence to support the study conclusions.
- The additional and updated figures are helpful.
- The methodology is explained in great detail.

Well done.

Reviewer #3 (Remarks to the Author):

I think the authors did a nice job of revising their manuscript and have addressed all of my concerns adequately. I think the edits to the methods regarding how the two variant stratum were handled was particularly helpful (in response to other Reviewer's comments). I have two very minor comments:

1. In Table S1, the central and pessimistic values for booster rate are the same. Was this a typo? I am assuming yes as the results presented for those two scenarios differ.
2. I'm curious why seeding was done exclusively through the 30-39 age group? Previous work (e.g., [10.1126/science.abe8372](https://doi.org/10.1126/science.abe8372)) suggests disproportionate transmission is through 20-49 year olds.

Response to reviewers

Reviewer #1 (Remarks to the Author):

I thank the author for their detailed and thoughtful responses to my queries. I will elaborate on my key questions below.

We are glad that our responses were well received by the reviewer, and have responded to their follow-up questions below.

Response to the question “Due to the small scale of the study and the topic, which has been explored at length for

Another reviewer also discussed the model's flexibility in other settings with limited data. Did the author also consider the scenario where the data stream quality is unevenly distributed (i.e., where one source has more significant details and size than other sources)? Would this introduce bias in the model output? Also, I understand the author used hospitalisation death data to prevent overestimating the number of deaths; however, due to the limited size of the data, the fitting may be relatively poor quality, especially when it is age-stratified. Could I understand more on how this could be addressed?

We have not considered the scenario where the data stream quality is uneven in terms of fitting to simulated data, but one data stream being much larger or having much greater detail than others could induce bias in the model estimates, especially if that datastream was daily case data, given variation in levels of testing and who was tested (symptomatic vs asymptomatic individuals) over time. This is why we explain in the last paragraph of the Discussion that relatively complete COVID-19 death/excess death or hospitalisation data would be essential for applying the model to another country, but case data would not be. Indeed, in early iterations of the analysis, we only fitted to the hospitalisation, death and seroprevalence data, and estimates of hospitalisations and deaths averted remained relatively stable when the case data was also included (but it also enabled estimation of the case reporting rate and better estimation of infection levels). In general, though, we would argue that inclusion of multiple datastreams in the inference can help to reduce the bias that could result from using a single datastream.

The right column of Fig. S4 on p. 10 of the Supplementary Information shows the fit of the model to the age-stratified hospital death data (while the left column shows the fit to the age-stratified hospitalisation data and Fig. S5 on p. 11 that to the age-stratified case data). It shows that the model fits the death data well, despite the low numbers of deaths in younger age groups. To allow sufficient flexibility in the model to fit the data

well, we estimated both the maximum (across age groups) probability of death given hospitalisation in each wave (to allow for changes in quality of hospital care over time) and the overdispersion parameter for the negative binomial observation process for hospital deaths in each age group (i.e. the amount of noise in the reporting of hospital deaths). Also fitting the probability of death given hospitalisation for each age group in the model might give an even closer fit to the age-stratified death data, but we did not judge this to be necessary given the closeness of the fit shown in Fig. S4.

Response to the question “It should also be noted that many countries experienced multiple waves of the same or” and “The impact of interventions also has received much attention as aforementioned. ...”

I would think the model may be hard to be generalised to other countries e.g. French Polynesia has a higher tourist-per-resident ratio than most other countries. The lockdown effect may have a more significant impact and be easier to enforce in this specific country than in other cities with high population densities.

It is important to distinguish between the model and inference framework and the results obtained with it for the setting of French Polynesia. Our intention is not to suggest, and indeed we believe we have not, that the results we have obtained with the model and inference framework for French Polynesia are generalisable to other settings (we agree with the reviewer that they are unlikely to be). However, we do believe that the framework we have developed addresses some deficiencies in existing modelling and inference frameworks for assessing vaccination and NPI impact for COVID-19, and thus could be applied to other settings to estimate intervention impact. The fundamental aspects of the model (the compartmental and strata structure, progression parameters, vaccine effectiveness parameters, etc.) and inference algorithm are not specific to French Polynesia, so with sufficient input data the framework could be used for other countries as described in the last paragraph of the Discussion on p. 7. As we intimate in the last sentence of the Discussion on p. 7, where we say that the framework could “provide valuable insights in settings with different vaccine and booster availability and NPI levels”, we would expect the results to be different in other settings. We also discuss the differences in estimated booster impact between our study and one for the UK (1) in the penultimate paragraph on p. 5 and highlight that this emphasises the need for analyses that account for country-specific factors in the abstract. Please also see our response to your next comment.

Response to the question “Whilst there is sufficient detail to repeat the exercise, the numerous assumptions would”

This may have been extensively discussed, but to add to it - I think the results from this paper, where the counterfactual example of no lockdown results in lower cases, will be hard to replicate in different countries. Lockdowns in principle can be very effective but their impacts have been heterogenous – they differ in type, compliance and duration. The relative remoteness and disconnectedness of Polynesia, as well as its population size, make it a relatively unique setting. I don't think this can be accounted for in the estimated values of the transmission rate parameters.

The counterfactual example of no lockdown actually results in slightly more cases overall between July 2020 and May 2022 (146,800 vs 145,000), but fewer hospitalisations (2891 vs 2937) and hospital deaths (356 vs 549), as stated in the second paragraph of the “Impact of NPIs” subsection of the Results on p. 3. As we note in the Discussion in the 2nd paragraph on p. 8, this result should be interpreted with caution as it is contingent on the assumption that hospitalised cases would have received the same level of care had hospitalisations been 75% higher during the first wave. Other studies have also noted non-linear effects in lockdown timing and subsequent dynamics, such as Figure 6 in Knock et al. (2), where it was estimated that an earlier UK lockdown in the first wave would – all else being equal – have led to a larger second wave, albeit with a smaller number of overall deaths.

Nevertheless, we agree with the reviewer that the result for French Polynesia is unlikely to be fully generalisable given heterogeneity in the implementation and impact of lockdowns between countries. We have therefore added the following sentence at the end of the penultimate paragraph on p. 6:

“Further, given the heterogeneity in type, compliance and duration of lockdowns between countries, and the relative uniqueness of French Polynesia in terms of remoteness and population size, this result is unlikely to be generalisable across countries.”

Overall comments:

It is hard to be convinced French Polynesia is a good microcosm for other countries. The model might be too complex to be replicated in other countries, and many assumptions are unrealistic in big cities. Nevertheless, this paper has produced many results and investigations on the effect of booster vaccination, cross-variant immunity and differing variant waves, which whilst they are key to important decision making, seem very tailored to French Polynesia. I would struggle to apply these methods to many of the islands in the Southeast Asia/South Pacific context.

We do not wish to claim that French Polynesia is necessarily a good microcosm for other countries. However, we would argue that the tailoring of the model to French Polynesia is actually relatively minimal and that it could be straightforwardly applied to other countries (especially Pacific island countries, but also countries with larger and more urban populations). The aspects of the model and parameterisation specific to French Polynesia (lockdown dates, surveillance and vaccination data, etc.) are all in the outer layer of the code and therefore easily modifiable. As in our previous responses, we would highlight that a similar model and framework based on the same underlying R simulation packages has been used to model the COVID-19 epidemic in England and estimate vaccination impact and variant severity and transmissibility (3,4), indicating that such an approach can be applied to other (larger) countries.

Reviewer #2 (Remarks to the Author):

The authors put in notable effort to reflect all feedback from the reviewers. It is commendable.

- The results reflect novel consideration of the role of immune status on intensity of waves. While the authors have made several simplifying assumptions, the modeling framework is very well documented now and the Discussion reflects on its flexibility for other settings, as well as the Limitations that might be overcome with additional data, for instance.
- The authors' clarification of language around the model fit and the addition of more discussion about how the model compares to other frameworks offers stronger evidence to support the study conclusions.
- The additional and updated figures are helpful.
- The methodology is explained in great detail.

Well done.

We are glad that the reviewer was happy with our revisions and thank them for their kind comments.

Reviewer #3 (Remarks to the Author):

I think the authors did a nice job of revising their manuscript and have addressed all of my concerns adequately. I think the edits to the methods regarding how the two variant stratum were handled was particularly helpful (in response to other Reviewer's comments). I have two very minor comments:

1. *In Table S1, the central and pessimistic values for booster rate are the same. Was this a typo? I am assuming yes as the results presented for those two scenarios differ.*

No, this was not a typo and we have added a footnote to the table to clarify this. We chose the central value of the booster waning rate to be the same as the pessimistic value to err on the side of being conservative about the estimated impact of boosters since there was limited data available with which to parametrise this aspect of the model. The difference in the results for the central and pessimistic parameter assumptions is due to the differences in the values of the other parameters in Table S1 (as the model was refitted and the counterfactual simulations rerun with the central and pessimistic parameter sets as a whole rather than individually). The influence of the booster waning rate on the estimated booster impact can be seen in the difference in the results of the “central” and “optimistic” parameter scenarios (since the biggest difference between these scenarios is in the booster waning rate).

2. *I'm curious why seeding was done exclusively through the 30-39 age group? Previous work (e.g., 10.1126/science.abe8372) suggests disproportionate transmission is through 20-49 year olds.*

Seeding was done exclusively in the 30-39 age group, since the age group into which the initial cases are seeded makes very little difference to the subsequent dynamics as the age distribution of infection quickly reaches that given by the dominant eigenvector of the next generation matrix (which is determined by the static contact matrix), especially in a deterministic setup, as we use here.

References

1. Barnard RC, Davies NG, Centre for Mathematical Modelling of Infectious Diseases COVID-19 working group, Jit M, Edmunds WJ. Modelling the medium-term dynamics of SARS-CoV-2 transmission in England in the Omicron era. *Nat Commun.* 2022 Aug 19;13(1):4879.
2. Knock ES, Whittles LK, Lees JA, Perez-Guzman PN, Verity R, FitzJohn RG, et al. Key epidemiological drivers and impact of interventions in the 2020 SARS-CoV-2 epidemic in England. *Sci Transl Med [Internet].* 2021 Jul 14;13(602). Available from: <http://dx.doi.org/10.1126/scitranslmed.abg4262>
3. Imai N, Rawson T, Knock ES, Sonabend R, Elmaci Y, Perez-Guzman PN, et al. Quantifying the effect of delaying the second COVID-19 vaccine dose in England: a mathematical modelling study. *Lancet Public Health.* 2023 Mar;8(3):e174–83.
4. Perez-Guzman PN, Knock E, Imai N, Rawson T, Elmaci Y, Alcada J, et al.

Epidemiological drivers of transmissibility and severity of SARS-CoV-2 in England.
Nat Commun. 2023 Jul 17;14(1):4279.